# Temporal-Aware Reasoning Optimization for Video Temporal Grounding

**Minghang Zheng** [1]  **Zihao Yin** [2]  **Yi Yang** [2]  **Yuxin Peng** [1]  **Yang Liu** [1 3]

## Abstract

Multi-modal Large Language Models (MLLMs) have achieved remarkable progress in video temporal grounding with reinforcement learning for generating reasoning paths. However, existing models often produce superficial reasoning, which offers limited guidance for precise temporal localization. This limitation stems from (1) inefficient random exploration and (2) reward functions that focus solely on the answer correctness while ignoring reasoning quality. To address these issues, we propose TaRO (Temporal-Aware Reasoning Optimization), a framework that explicitly enhances the model's ability of thinking with time. First, we introduce a Constructive Reasoning Exploration that leverages pre-generated dense captions to construct reasoning paths grounded in explicit visual cues and timestamps, enabling efficient exploration of high-quality time-aware reasoning. Second, to evaluate reasoning quality, we design a Temporal-Sensitivity Reward. High-quality reasoning should be anchored to specific events and timestamps. If the event boundary under thinking is disrupted, such reasoning should become invalid, leading to a drop in the logit of the reasoning path. We utilize this drop as a critique of reasoning quality. Finally, TaRO follows a progressive curriculum, which starts by utilizing this reward to select better constructed reasoning paths, and evolves to a free exploration phase where the model autonomously generates effective reasoning. Experiments demonstrate that TaRO achieves state-of-the-art performance on VTG benchmarks. Code is available at https://github.com/oceanflowlab/TaRO.

[1]Wangxuan Institute of Computer Technology, Peking University [2]Central Media Technology Institute, Huawei Technologies Ltd. [3]State Key Laboratory of General Artificial Intelligence, Peking University. Correspondence to: Yang Liu <yangliu@pku.edu.cn>.

*Proceedings of the 43rd International Conference on Machine Learning*, Seoul, South Korea. PMLR 306, 2026. Copyright 2026 by the author(s).

## 1. Introduction

Video temporal grounding (VTG) aims to localize the precise temporal segment in an untrimmed video that corresponds to a given natural language query. Recent advances in multimodal large language models (MLLMs) have significantly expanded the capacity of models to reason over videos and complex queries. In particular, reinforcement learning (RL) based approaches have emerged as a promising direction, as they allow models to generate reasoning paths that guide temporal localization.

Despite these developments, current RL-based methods (Wang et al., 2026; Yan et al., 2026) still struggle to produce reasoning that is genuinely useful for grounding. They often generate superficial descriptions that fail to identify the specific video evidence required for the answer. This issue is evident in our analysis of Time-R1 (Wang et al., 2026), a state-of-the-art model explicitly trained to perform reasoning for VTG using RL. As shown in Fig. 1(a), we compare Time-R1's performance under two RL settings: one with reasoning chains for both training and inference, and another with direct answer output for both training and inference. As we can see, the performance is comparable between the two settings. This suggests that although Time-R1 is trained to reason, the generated reasoning contributes little to the final grounding. We argue that this stems from two main limitations in its RL paradigms. *(1) Firstly, the random rollout during RL blindly explores the vast reasoning space of videos without guidance.* Consequently, the model predominantly explores low-quality trajectories, often leading to suboptimal and superficial reasoning. *(2) Secondly, current reward designs primarily focus on the correctness of the final answer (e.g., IoU) while ignoring the quality of the reasoning process itself.* As a result, reasoning paths that do not genuinely depend on visual-temporal evidence may still be reinforced, leading to rely on spurious correlations and hindering the ability to generalize to zero-shot scenarios.

To overcome these challenges, we propose a Temporal-Aware Reasoning Optimization (TaRO) framework that explicitly encourages models to *think with time*. As shown in Fig. 1 (b), we define effective reasoning in VTG as the ability to selectively attend to critical visual cues and be temporally sensitive, anchoring these cues to specific timestamps. *First, to enable the model to identify critical visual*

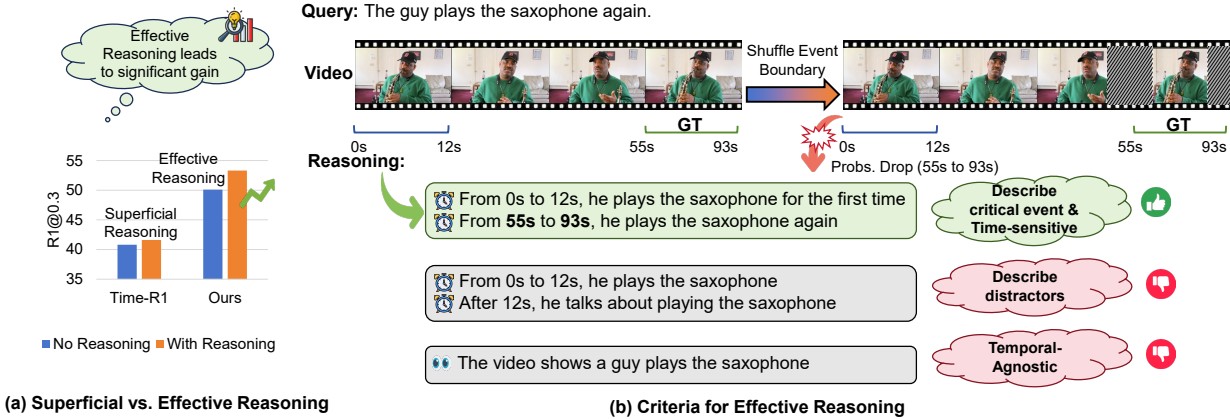

*Figure 1.* **(a) Comparison of performance gains of different reasoning.** Models are trained using RL with reasoning chains or direct answer output for both training and inference. **(b) Our criterion for reasoning quality.** Effective reasoning should selectively attend to critical visual cues and be temporally sensitive, anchoring these cues to specific timestamps.

*cues and adopt the thinking with time paradigm, we propose a Constructive Reasoning Exploration strategy.* Instead of relying on random exploration, we construct high-quality reasoning trajectories by leveraging dense video captions with explicit timestamps. Since describing every event in the video introduces noise and distractors and reasoning redundancy can actually lower performance (Wu et al., 2025; Hwang et al., 2025), we only sample some captions and concatenate them in chronological order to form reasoning trajectories. The model then completes the reasoning and produces a final grounding prediction, yielding a rollout. Since different sampled combinations lead to reasoning processes of varying quality, the model is encouraged to learn which captions are critical for grounding and which are distracting through the reward supervision in RL. *Second, to ensure the reasoning is strictly grounded in the correct visual segments, we design a Temporal-Sensitivity Reward.* We achieve this by locally shuffling frames near the ground-truth start and end timestamps. The intuition is that if the reasoning accurately describes the events and timestamps occurring at these critical moments, disrupting the temporal order of key frames will render the generated reasoning invalid, thereby causing a significant drop in the confidence of the reasoning. We utilize this confidence drop as a reward signal, explicitly forcing the model to generate reasoning that is tightly coupled with the specific visual evidence at the correct timestamps. *Finally, TaRO follows a Progressive Curriculum.* We supervise the model using constructed rollouts in the early stage to teach the model which visual cues are worth attending to and establish a paradigm for thinking with time. In the later stage, we transition to traditional RL with random rollouts, allowing the model to autonomously refine its reasoning strategies under the guidance of our Temporal-Sensitivity Reward and standard IoU rewards.

Our contributions are summarized as follows: (1) We pro-

pose TaRO, a novel framework that enhances the reasoning capabilities of MLLMs for VTG by explicitly encouraging thinking with time. (2) We introduce a Constructive Reasoning Exploration to teach the model which visual cues are worth attending to, and a Temporal-Sensitivity Reward to evaluate the quality of reasoning and encourage reasoning anchored in critical events and timestamps. (3) Extensive experiments on VTG benchmarks demonstrate that TaRO achieves state-of-the-art performance.

## 2. Related Work

### 2.1. Video Temporal Grounding

Traditional approaches can be categorized into proposal-based (Gao et al., 2017; Xiao et al., 2021; Liu et al., 2021; 2026; Zheng et al., 2025a;b) and proposal-free methods (Yuan et al., 2019; Mun et al., 2020; Zhang et al., 2020; Pan et al., 2023; Mu et al., 2024). While effective in closed domains, these methods have a poor ability to generalize to unseen scenarios. With the advent of MLLMs (Yuxin et al., 2026; Zheng et al., 2026), methods like TimeChat (Ren et al., 2024), UniVTG (Lin et al., 2023), and ChatVTG (Qu et al., 2024) formulate grounding as a text generation task, outputting timestamps directly using MLLMs. To better capture temporal information, UniTime (Li et al., 2026b) and DisTime (Zeng et al., 2025b) introduce specialized time tokens or distribution-based representations. More recently, RL has been introduced to the VTG domain. For example, VideoChat-R1.5 (Yan et al., 2026) and Time-R1 (Wang et al., 2026) propose a reasoning-guided post-training framework that encourages models to generate a thinking process before prediction and achieve state-of-the-art performance. Despite these advances, existing RL-based VTG methods still face challenges. First, they rely on random rollout during RL, which blindly explores the vast reasoning space of videos,

often leading to suboptimal and superficial reasoning policies. Second, their reward mechanisms predominantly focus on the correctness of the final answer (e.g., IoU) instead of ensuring the reasoning is genuinely grounded in critical events and timestamps. This may lead models to rely on spurious correlations and hinder the ability to generalize to zero-shot scenarios. Our TaRO framework addresses these issues by introducing a Constructive Reasoning Exploration to teach the model which visual cues are worth attending to, and a Temporal-Sensitivity Reward to evaluate the quality of reasoning and encourage reasoning anchored in critical events and timestamps. To our knowledge, we are the first in VTG to explicitly encourage MLLMs to think with time.

### 2.2. MLLM Reasoning in Videos QA

Recently, the reasoning capabilities of MLLMs have been developing rapidly. (Lyu et al., 2025; Yang et al., 2025b; Yin et al., 2025; Yang et al., 2025a; Mo et al., 2025) For example, DeepVideo-R1 (Park et al., 2026) applies RL fine-tuning to enhance the video reasoning for video QA. One-Thinker (Feng et al., 2025a) proposes an all-in-one reasoning capable of handling diverse visual tasks. Video-R1 (Feng et al., 2026) introduces T-GRPO to encourage temporal awareness by contrasting the average QA performance of a group of responses on ordered versus shuffled entire videos. However, this remains a group-level, answer-oriented reward: it assigns a uniform score to all responses based on final accuracy, failing to assess the quality of individual reasoning. Moreover, this strategy is ill-suited for VTG. Shuffling the entire video renders the temporal ground truth invalid. Consequently, it becomes impossible to calculate the answer accuracy (e.g., IoU) on the shuffled video, which is a prerequisite for their reward calculation. In the context of VTG, where the goal is to output precise time intervals, the model requires a stricter ability to *think with time*. To close this gap, TaRO introduces an instance-level Temporal-Sensitive Reward. Unlike Video-R1's group-level, answer-oriented reward, our reward directly measures the temporal-sensitivity of each specific reasoning path, ensuring the model is rewarded only when its reasoning process is anchored in critical timestamps.

## 3. Method

**Problem Formulation.** Given an untrimmed video $V$ and a natural language query $Q$, the goal of VTG is to predict a temporal segment $y = (t^s, t^e)$, where $t^s$ and $t^e$ denote the start and end timestamps of the target event.

### 3.1. Revisiting RL in VTG

Recent reinforcement learning (RL) based approaches, such as Time-R1 (Wang et al., 2026), involve an MLLM acting as the policy model $\pi_\theta$, which takes the video and query as

input and generates a response sequence $o$. To explicitly facilitate reasoning, the model is prompted to structure its output in a Chain-of-Thought (CoT) format, consisting of an intermediate reasoning trace $r$ followed by a final timestamp prediction, denoted as $o$ =`<think>`$r$`</think>` `<answer>`$\hat{t}^s$ `to` $\hat{t}^e$`</answer>`.

Time-R1 uses the GRPO (Shao et al., 2024) algorithm for policy optimization. During training, for a given input $(V, Q)$, the MLLM generates a group of $G$ rollouts $\{o_1, o_2, \ldots, o_G\}$ via random sampling. The optimization is driven by a composite reward function $r(o)$ that evaluates the quality of each sampled response, which is the sum of a format reward and a IoU reward:

$$r(o) = r_{\text{form}}(o) + r_{\text{tIoU}}(o). \tag{1}$$

The format reward $r_{\text{form}}(o)$ is a binary indicator that encourages the model to strictly follow the predefined structural template. The IoU reward $r_{\text{tIoU}}(o)$ evaluates the accuracy of the predicted segment $[\hat{t}^s, \hat{t}^e]$ against the ground truth $[t^s, t^e]$. With the computed rewards for each rollout, the model is optimized using GRPO (Shao et al., 2024) to maximize the expected cumulative reward.

However, it faces two limitations. Firstly, the random rollout strategy fails to genuinely *think with time*. Our statistical analysis on the Charades-STA test set reveals that only 8.3% of Time-R1's generated reasoning contain explicit timestamps. Consequently, the explored reasoning paths are often superficial and offer little substantive guidance for temporal grounding. Secondly, the reward function $r(o)$ assesses only the correctness of the final timestamp $[t_s, t_e]$ and the superficial format of the output, ignoring evaluating the quality of the intermediate reasoning process $r$.

### 3.2. Overview

To overcome the limitations, we propose **TaRO** (**T**emporal-**A**ware **R**easoning **O**ptimization). As illustrated in Fig. 2, TaRO is designed to explicitly encourage the model to *think with time* by anchoring its reasoning process in concrete visual-temporal evidence. Directly exploring the vast reasoning space of videos via random rollouts is inefficient. Firstly, to provide high-quality initial guidance, we propose a **Constructive Reasoning Exploration** shown in the top block of Fig. 2. We first employ an off-the-shelf captioner to generate dense captions for the input video, obtaining a set of atomic events with precise timestamps. Then we randomly select a subset of these events and concatenate them chronologically. These sampled captions are wrapped into the standard thinking format to form constructed reasoning traces. Secondly, to evaluate the quality of reasoning traces, introduce a **Temporal-Sensitive Reward** as shown in the bottom block of Fig. 2. The core intuition is that high-quality reasoning should rely on critical events and

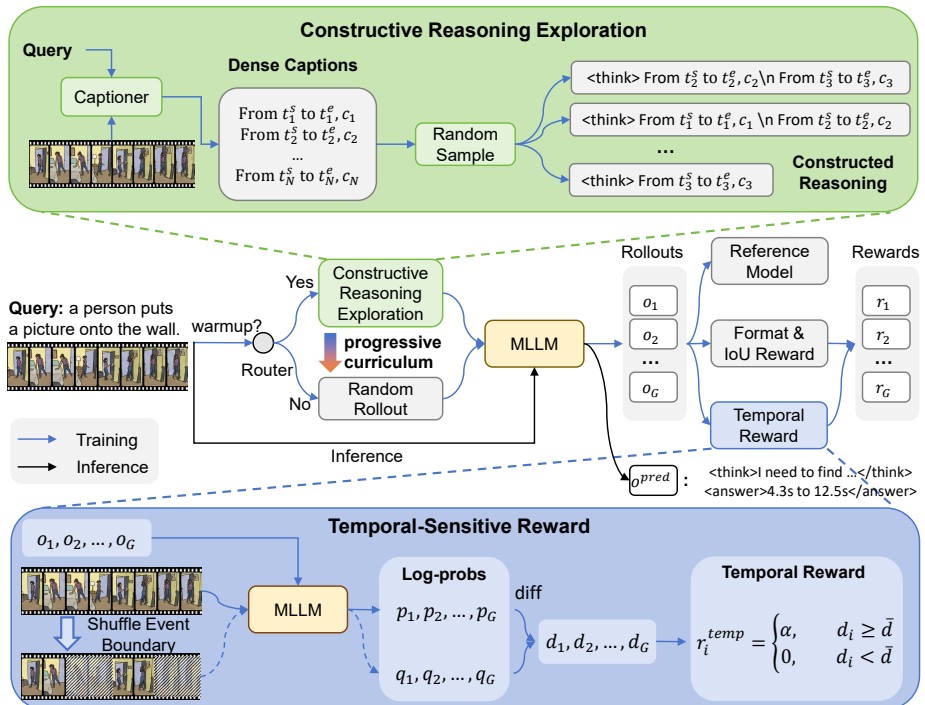

*Figure 2.* **Pipeline of our TaRO methods.** We propose two components to enable the model to *think with time*: **Constructive Reasoning Exploration** utilizes dense captions to construct informative reasoning traces for high-quality initialization; **Temporal-Sensitive Reward** evaluates reasoning quality by measuring the model's sensitivity to ground-truth event boundaries. These components are integrated via a **Progressive Curriculum** that transitions the model from guided learning to autonomous exploration.

timestamps in the video. Therefore, for a generated rollout containing reasoning $o_i$, we perturb the video by shuffling frames near the ground-truth event boundaries and compute the log-probabilities of the reasoning tokens given the original video and the shuffled video. The difference between these log-probabilities quantifies the model's sensitivity to temporal information. A significant drop indicates that the reasoning relies on critical temporal cues, triggering a positive reward. Thirdly, to effectively integrate the guided exploration and the intrinsic reasoning capability of the model, we adopt a **Progressive Curriculum** learning strategy. In the warmup training stage, the model is trained with the Constructive Reasoning Exploration rollouts to quickly learn to identify and utilize informative temporal captions. As training progresses, we transition to standard random rollouts, allowing the model to freely explore and refine its reasoning strategy autonomously under the guidance of the Temporal-Sensitivity Reward.

### 3.3. Constructive Reasoning Exploration

To provide high-quality initial guidance, we propose a Constructive Reasoning Exploration strategy.

**Reasoning Construction.** A good reasoning process should ground on concrete events and timestamps. To obtain potential events in the video, we utilize Gemini-

3-Pro as a captioner to generate a set of dense captions $\mathcal{C} = \{(t_k^s, t_k^e, c_k)\}_{k=1}^N$. Each caption $c_k$ describes an event occurring between timestamps $t_k^s$ and $t_k^e$. The specific prompt used for caption generation is provided in the Appendix B. Instead of generating $G$ rollouts using the MLLM from scratch, we construct rollouts by sampling from these captions. As not all captions are relevant to the query and research (Wu et al., 2025; Hwang et al., 2025) shows that reasoning redundancy can actually lower accuracy, so we only randomly sample a subset of captions $\hat{\mathcal{C}}_i \subset \mathcal{C}$ and sort them chronologically for each rollout $i \in \{1, \ldots, G\}$. These captions are formatted into a reasoning trace $r_i^{\text{cons}}$ following the template: `<think>From` $t_{j_1}^s$ `to` $t_{j_1}^e$`,` $c_{j_1}$ `...From` $t_{j_m}^s$ `to` $t_{j_m}^e$`,` $c_{j_m}$, where $j_1, \ldots, j_m$ is the captions indexes in $\hat{\mathcal{C}}_i$. This constructed reasoning is then fed into the MLLM, which is prompted to continue the generation, completing any remaining reasoning and predicting the final answer to form the full output $o_i$. This process effectively creates a distribution of reasoning paths that are semantically grounded in the video, acting as a warm-up for the model's reasoning capabilities.

**Advantage-Weighted Behavioral Cloning Loss.** Since the reasoning components of these rollouts are constructed externally rather than generated by the current policy $\pi_\theta$, they constitute *off-policy* data. Consequently, standard on-policy

gradient methods like GRPO are not directly applicable. To solve this problem, we use the Advantage-Weighted Behavioral Cloning Loss (AW-BC), which treats this phase as an imitation learning (Zare et al., 2024) process. We aim to encourage the model to imitate the constructed reasoning paths that lead to higher rewards. We first calculate the reward $r(o_i)$ for each constructed rollout, which will be further introduced in Sec. 3.4. To determine the quality of each rollout relative to the group, we compute the advantage $A_i$ using the group computation (Shao et al., 2024): $A_i = \frac{r(o_i) - \mu_r}{\sigma_r}$, where $\mu_r$ and $\sigma_r$ are the mean and standard deviation of rewards within the group. A positive advantage ($A_i > 0$) indicates that the specific combination of sampled captions in $o_i$ provided effective cues for localization. We then employ an advantage-weighted Behavioral Cloning objective (Torabi et al., 2018), effectively upweighting the likelihood of high-quality rollouts in the optimization:

$$\mathcal{L}_{AW-BC} = -\frac{1}{G} \sum_{i=1}^{G} \mathbb{I}(A_i > 0) \cdot A_i \cdot \log \pi_\theta(o_i | V, Q). \quad (2)$$

Here, $\mathbb{I}(\cdot)$ is the indicator function. This loss ensures that the model selectively learns from constructed examples that empirically lead to higher reward.

**Discussion.**

The core objective of our method is to empower the model to *think with time* and selectively identify visual content that assists in grounding queried events. To achieve this, one approach is to employ SFT on static CoT data as a cold start. However, our proposed pipeline offers two distinct advantages over this conventional method. First, studies (Wang et al., 2026) have shown that SFT is ill-suited for supervising continuous temporal output. For example, a prediction of 3.0s is semantically nearly identical to a ground truth of 2.9s, yet SFT penalizes this token mismatch heavily. Instead, our RL-based objective, driven by the reward (e.g., IoU), tolerates small numerical deviations. Second, SFT relies on static data demonstrations, teaching the model how to mimic a fixed reasoning path. In contrast, our method generates dynamic training data. By randomly sampling caption combinations and prompting the model to autonomously complete the subsequent reasoning and prediction, we expose the model to diverse reasoning variations. Through the reward, the model explicitly learns to distinguish between informative visual cues (high reward) and distractors (low reward).

### 3.4. Temporal-Sensitive Reward

To ensure that the generated reasoning $r$ is effectively grounded on critical timestamps, we introduce a Temporal-Sensitive Reward that verifies the dependency of the reasoning path on the specific temporal information of events.

**Sensitivity Measurement.** The core intuition is that if the reasoning accurately describes the specific events and timestamps occurring at critical moments, disrupting the temporal order of these key frames will render the generated reasoning invalid, thereby causing a significant drop in the confidence of the reasoning. Based on this, for a generated rollout $o_i$ containing reasoning trace $r_i$, we first compute the average log-probability of the reasoning tokens conditioned on the original video $V$ and the query $Q$:

$$p_i = \frac{1}{|r_i|} \sum_{k=1}^{|r_i|} \log \pi(r_{i,k} | V, Q, r_{i,<k}), \quad (3)$$

where $|r_i|$ denotes the length of the reasoning sequence. Next, we construct a perturbed video $V'$ to test the model's awareness of event boundaries. We randomly shuffle the frames within a small temporal window $\Delta t$ centered around the ground-truth start timestamp $t^s$ and end timestamp $t^e$, while keeping the rest of the video unchanged. We then evaluate the log-probability of generating the same reasoning trace $r_i$ conditioned on this perturbed video $V'$:

$$q_i = \frac{1}{|r_i|} \sum_{k=1}^{|r_i|} \log \pi(r_{i,k} | V', Q, r_{i,<k}). \quad (4)$$

The temporal sensitivity score $d_i$ is defined as the drop in confidence between the original and perturbed conditions:

$$d_i = p_i - q_i. \quad (5)$$

A larger positive $d_i$ indicates that the reasoning is strongly anchored to the correct visual-temporal evidence of the critical frames, as the model finds the reasoning much less plausible when those frames are disordered.

**Reward Formulation.** To compute the reward, we first calculate the baseline sensitivity $\bar{d}$ as the average score within the current group of $G$ rollouts, i.e., $\bar{d} = \frac{1}{G} \sum_{j=1}^{G} d_j$. Reasoning paths that exhibit higher sensitivity than the group average are considered to be better grounded. Thus, we define the temporal-sensitive reward as:

$$r_i^{\text{temp}} = \begin{cases} \alpha, & \text{if } d_i > \bar{d} \\ 0, & \text{otherwise} \end{cases} \quad (6)$$

where $\alpha$ is a hyperparameter controlling the reward magnitude. In the early stages of training, or when the model's predicted answer is completely incorrect, the reasoning is likely flawed regardless of its sensitivity (e.g., confidently describing the wrong event). To prevent the model from overfitting to the temporal reward while neglecting the primary grounding task, we employ a gating mechanism based on the grounding accuracy. The temporal reward is only applied when the IoU between the predicted and ground-truth

segments exceeds a threshold $\tau$. The final composite reward for the reinforcement phase is formulated as:

$$r(o_i) = r_{\text{form}}(o_i) + r_{\text{tIoU}}(o_i) + r_i^{\text{temp}}\mathbb{I}(\text{IoU}_i > \tau), \quad (7)$$

where $\mathbb{I}(\cdot)$ is the indicator function.

### 3.5. Progressive Curriculum

To effectively bridge the gap between using constructed reasoning and autonomously generating robust reasoning, we implement a progressive curriculum learning strategy.

**Warm-up with Constructive Reasoning.** In the initial phase, the primary goal is to warm up the model, specifically to initialize its capability to selectively attend to critical sub-events and reason with explicit timestamps. By utilizing the Constructive Reasoning Exploration strategy, we teach the model which visual cues to select and how to ground them temporally. The model is then optimized by behavioral cloning using $\mathcal{L}_{AW-BC}$ (Eq. 2).

**Self-Exploration** Once the model has acquired basic temporal reasoning capabilities, we transition to the second phase to enable autonomous exploration to create new reasoning strategies. In this stage, we switch the data flow to the random rollout. The model generates its own reasoning and answers $o_i \sim \pi_\theta(o|V, Q)$ without external construction. We optimize the model using GRPO, with the difference that the original reward function (Eq. 1) is replaced by our Temporal-Sensitive Reward (Eq. 7).

This two-stage curriculum ensures a smooth transition from supervised imitation to autonomous creation of new reasoning strategies. After training, 100% of the reasoning traces generated by our model contain explicit timestamps on the Charades-STA test set (only 8.3% for Time-R1), indicating our model has acquired the capability to *think with time*.

## 4. Experiments

### 4.1. Datasets and Evaluation

**Training Data.** To ensure a fair comparison, we utilize the exact same training set as Time-R1 (Wang et al., 2026), containing 2,500 samples sampled from existing datasets, including YT-Temporal (Zellers et al., 2022), DiDeMo (Anne Hendricks et al., 2017), QuerYD (Oncescu et al., 2021), InternVid (Wang et al., 2024a), and HowTo100M (Miech et al., 2019). This specifically allows us to isolate the contributions of our reasoning framework and optimization strategy from the training data.

**Evaluation Benchmarks.** We assess the zero-shot performance of TaRO on four established benchmarks. (1) *Charades-STA* (Gao et al., 2017) comprises videos recording daily indoor activities. (2) *ActivityNet Captions* (Krishna et al., 2017) is a large-scale benchmark containing 20k

untrimmed videos focusing on complex human activities. (3) *QVHighlights* (Lei et al., 2021) features a diverse collection of content ranging from lifestyle vlogs to news reports. (4) *TVGBench* (Wang et al., 2026) is a recently proposed comprehensive benchmark designed to rigorously evaluate temporal grounding capabilities across diverse query types. To evaluate our method's scalability to long videos, we further evaluate the zero-shot performance on two long-form datasets: TACoS (Regneri et al., 2013) and Ego4D NLQ (Xu et al., 2022).

**Evaluation Metrics.** Following standard practices in the VTG (Wang et al., 2026; Li et al., 2026b), we employ R1@$m$ as our primary evaluation metric, which calculates the percentage of samples where the predicted segment has an IoU greater than $m \in \{0.3, 0.5, 0.7\}$.

### 4.2. Implementation Details

Unless otherwise stated, we use Qwen2.5-VL-7B (Bai et al., 2025b) as the base model for fair comparison with top-performance methods (Wang et al., 2026; Li et al., 2026b; Yan et al., 2026). More details can be found in Appendix B.

### 4.3. Comparison with State-of-the-Art Methods

**Zero-shot Performance.** We evaluate the zero-shot performance of TaRO against state-of-the-art methods. We also tested using Qwen2.5-VL-3B-Instruct and Qwen3-VL-8B-Instruct to demonstrate that our method is effective across MLLMs of different sizes and architectures. The results are shown in Tab. 1. (1) As we can see, when using Qwen2.5-VL-7B-Instruct, TaRO consistently outperforms existing methods across all four benchmarks. Notably, compared to the recent RL-based method Time-R1, TaRO achieves significant improvements. For example, on the R1@0.5 metric, we outperform Time-R1 by 4.0% on Charades-STA, 0.8% on ActivityNet, 3.2% on QvHighlights, and 8.4% on TVGBench. This demonstrates that our model's reasoning, that ground on key visual cues and specific timestamps, is effective for improving its zero-shot generalization. (2) Our TaRO achieves stable performance improvements on models of different sizes and architectures. Notably, the performance gains over Time-R1 are more pronounced on the smaller Qwen2.5-VL-3B model, where TaRO achieves a 6.7% improvement in R1@0.5 on ActivityNet and 23.4% on QVHighlights. We attribute this to the fact that smaller models possess weaker initial reasoning and grounding capabilities, making it more difficult for standard random rollout strategies to discover high-quality reasoning trajectories. In contrast, our Constructive Reasoning Exploration and Temporal-Sensitive Reward provides high-quality reasoning examples and explicit reward guidance for better initialization. On the Qwen3-VL-8B architecture, TaRO also exhibits superior grounding capabilities compared to both the base

*Table 1.* **Zero-shot performance comparison with multimodal large language models on video temporal grounding benchmarks.**

| Method | Size | Charades-STA (Gao et al., 2017) | | | ActivityNet (Krishna et al., 2017) | | | QVHighlights (Lei et al., 2021) | | | TVGBench (Wang et al., 2026) | | |
|---|---|---|---|---|---|---|---|---|---|---|---|---|---|
| | | R1@0.3 | R1@0.5 | R1@0.7 | R1@0.3 | R1@0.5 | R1@0.7 | R1@0.3 | R1@0.5 | R1@0.7 | R1@0.3 | R1@0.5 | R1@0.7 |
| ChatVTG (Qu et al., 2024) | 7B | 52.7 | 33.0 | 15.9 | 40.7 | 22.5 | 9.4 | - | - | - | - | - | - |
| TimeChat (Ren et al., 2024) | 7B | - | 32.2 | 13.4 | 36.2 | 20.2 | 9.5 | - | 8.32 | 4.26 | 22.4 | 11.9 | 5.3 |
| HawkEye (Wang et al., 2024b) | 7B | 50.6 | 31.4 | 14.5 | 49.1 | 29.3 | 10.7 | - | - | - | - | - | - |
| VTimeLLM (Huang et al., 2024) | 7B | 51.0 | 27.5 | 11.4 | 44.0 | 27.8 | 14.3 | - | 26.1 | 11.1 | - | - | - |
| TimeSuite (Zeng et al., 2025a) | 7B | 69.9 | 48.7 | 24.0 | - | 16.6 | 9.28 | - | 12.3 | 9.16 | 31.1 | 18.0 | 8.9 |
| VideoChat-Flash (Li et al., 2026a) | 7B | 74.5 | 53.1 | 27.6 | - | - | - | - | - | - | 32.8 | 19.8 | 10.4 |
| TRACE (Guo et al., 2025) | 7B | - | 40.3 | 19.4 | - | - | - | - | - | - | 37.0 | 25.5 | 14.6 |
| *Qwen2.5-VL-7B-Instruct as base model* | | | | | | | | | | | | | |
| Qwen2.5-VL-7B-Instruct (Bai et al., 2025b) | 7B | 72.5 | 53.6 | 28.5 | 24.4 | 13.6 | 6.7 | 15.93 | 7.10 | 4.19 | 35.3 | 20.0 | 12.5 |
| UniTime (Li et al., 2026b) | 7B | - | 59.1 | 31.9 | - | 22.8 | 14.1 | - | 41.0 | 31.5 | - | - | - |
| VideoChat-R1.5 (Yan et al., 2026) | 7B | - | - | - | 52.4 | 32.3 | 16.8 | 71.4 | 55.8 | 38.4 | - | - | - |
| Time-R1 (Wang et al., 2026) | 7B | 78.1 | 60.8 | 35.3 | 58.6 | 39.0 | 21.4 | 80.3 | 66.2 | 44.8 | 41.8 | 29.4 | 16.4 |
| **TaRO (Ours)** | 7B | **79.7** | **64.8** | **38.4** | **60.6** | **39.8** | **21.4** | **82.6** | **69.4** | **48.8** | **54.6** | **37.8** | **20.0** |
| *Qwen2.5-VL-3B-Instruct as base model* | | | | | | | | | | | | | |
| Qwen2.5-VL-3B-Instruct (Bai et al., 2025b) | 3B | 62.1 | 42.0 | 20.1 | 26.5 | 15.2 | 7.2 | 16.8 | 9.9 | 3.1 | 26.1 | 17.6 | 10.3 |
| Time-R1 (Wang et al., 2026) | 3B | 74.6 | 53.1 | 26.0 | 46.2 | 26.0 | 11.4 | 40.8 | 19.7 | 5.9 | 40.1 | 24.4 | 11.8 |
| **TaRO (Ours)** | 3B | **75.1** | **55.2** | **28.6** | **52.3** | **32.7** | **14.2** | **65.6** | **43.1** | **20.8** | **44.3** | **28.0** | **13.4** |
| *Qwen3-VL-8B-Instruct as base model* | | | | | | | | | | | | | |
| Qwen3-VL-8B-Instruct (Bai et al., 2025a) | 8B | 72.7 | 49.8 | 21.3 | 45.9 | 31.7 | 19.8 | 57.2 | 45.9 | 34.5 | 37.3 | 26.5 | 15.0 |
| Qwen3-VL-8B-Think (Bai et al., 2025a) | 8B | 72.7 | 57.9 | 32.6 | 44.0 | 30.5 | 19.2 | 69.4 | 56.3 | 42.7 | 38.1 | 25.9 | 13.7 |
| **TaRO (Ours)** | 8B | **83.2** | **67.8** | **41.9** | **59.7** | **40.6** | **23.9** | **82.7** | **68.5** | **51.7** | **54.3** | **39.1** | **20.3** |

*Table 2.* **Finetuned performance with different data amounts.**

| Method | Data Ratio | ActivityNet (Krishna et al., 2017) | | |
|---|---|---|---|---|
| | | R1@0.3 | R1@0.5 | R1@0.7 |
| Mr.BLIP (Meinardus et al., 2024) | 100% | - | 53.9 | 35.6 |
| UniTime (Li et al., 2026b) | | - | 54.8 | **36.6** |
| OneThinker (Feng et al., 2025b) | | 65.0 | 43.6 | 25.7 |
| Time-R1 (Wang et al., 2026) | | 73.3 | 55.6 | 34.0 |
| | 10% | 70.9 | 53.4 | 31.8 |
| Time-R1 (Wang et al., 2026) | 30% | 71.6 | 54.1 | 31.1 |
| | 50% | 72.2 | 56.1 | 32.9 |
| | 10% | 72.0 | 54.2 | 32.5 |
| **TaRO (Ours)** | 30% | 73.0 | 55.3 | 33.2 |
| | 50% | **74.6** | **57.2** | 34.8 |

*Table 3.* **Zero-shot performance comparison with multimodal large language models on long-form video benchmarks.**

| Method | TACoS (Regneri et al., 2013) | | | Ego4D NLQ (Xu et al., 2022) | | |
|---|---|---|---|---|---|---|
| | R1@0.3 | R1@0.5 | R1@0.7 | R1@0.3 | R1@0.5 | R1@0.7 |
| *Qwen2.5-VL-7B-Instruct as base model* | | | | | | |
| VideoChat-R1.5 (Yan et al., 2026) | 28.6 | 16.2 | 6.17 | 3.84 | 1.78 | 0.86 |
| Time-R1 (Wang et al., 2026) | 36.8 | 21.5 | 7.90 | 6.48 | 3.29 | 1.49 |
| **TaRO (Ours)** | **37.9** | **22.5** | **8.60** | **8.19** | **4.55** | **1.93** |
| *Qwen3-VL-8B-Instruct as base model* | | | | | | |
| Qwen3-VL-8B-Instruct | 38.6 | 30.5 | 20.4 | 5.10 | 2.94 | 1.52 |
| **TaRO (Ours)** | **52.3** | **36.5** | **22.5** | **13.80** | **7.86** | **4.17** |

We additionally compare with the latest Qwen3-VL-8B, which has shown better performance in long-videos than Qwen2.5-VL. Built on top, our method achieves significant improvements (e.g., +13.7% on TACoS and +8.7% on Ego4D NLQ for R1@0.3). This demonstrates that our method is scalable to long-form videos and generalizable to new, stronger baseline architectures.

### 4.4. Ablation Studies

In this section, we conduct extensive ablation experiments to verify the effectiveness of each component in TaRO. We use Qwen2.5-VL-7B-Instruct as the base model and report the zero-shot performance on the Charades-STA dataset.

*Table 4.* **Effectiveness of individual components in TaRO.**

| CRE | TR | PC | Charades-STA | | |
|---|---|---|---|---|---|
| | | | R1@0.3 | R1@0.5 | R1@0.7 |
| × | × | × | 78.2 | 61.1 | 35.2 |
| × | ✓ | × | 78.6 | 63.1 | 36.1 |
| ✓ | × | × | 70.4 | 51.6 | 25.5 |
| ✓ | ✓ | × | 71.1 | 53.0 | 26.3 |
| ✓ | × | ✓ | 78.9 | 63.9 | 36.3 |
| ✓ | ✓ | ✓ | **79.7** | **64.8** | **38.4** |

**Component Effectiveness.** We first investigate the con-

instruction model and the thinking-enhanced baseline.

**Data-Efficient Finetuning.** One advantage of TaRO is that once the model establishes a robust *thinking with time* paradigm, it requires fewer data to adapt to specific downstream distributions. We investigate the data efficiency of TaRO by finetuning on the ActivityNet training set with limited data ratios. As shown in Table 2, with only **10%** of the data for finetuning, TaRO achieves a significant performance leap over the zero-shot setting, boosting R1@0.5 from 39.8% to 54.2% and achieving comparable performance to state-of-the-art methods that utilize 100% of the training data. In addition, across all ratios, our method consistently outperforms Time-R1, confirming the data efficiency of our approach.

**Performance on long-form videos.** To evaluate our method's scalability to long videos, we further evaluate the zero-shot performance on two long-form datasets: TACoS (Regneri et al., 2013) (avg. 367s) and Ego4D NLQ (Xu et al., 2022) (avg. 499s). As we can see, using the same Qwen2.5-VL-7B, our method still achieves SOTA and outperforms the second-best baseline, Time-R1.

*Table 5.* **Ablation of Constructive Reasoning Exploration.**

| Strategy | Charades-STA | | |
|---|---|---|---|
| | R1@0.3 | R1@0.5 | R1@0.7 |
| SFT | 78.2 | 63.3 | 35.9 |
| CRE w/ All Captions | 78.4 | 63.6 | 36.6 |
| CRE w/ Sample Captions | **79.7** | **64.8** | **38.4** |
| GRPO Loss | 63.5 | 46.7 | 22.1 |
| AW-BC Loss (No Weighting) | 78.5 | 63.0 | 37.1 |
| AW-BC Loss (No $\mathbb{I}(A > 0)$ Filter) | 78.9 | 63.1 | 33.7 |
| AW-BC Loss | **79.7** | **64.8** | **38.4** |

*Table 6.* **Ablation of different external dense captions.**

| Strategy | Charades-STA | | |
|---|---|---|---|
| | R1@0.3 | R1@0.5 | R1@0.7 |
| Baseline | 78.1 | 60.8 | 35.3 |
| Ours w/o CRE | 78.6 | 63.1 | 36.1 |
| Ours w/ CRE (Gemini-3-Pro) | 79.7 | 64.8 | 38.4 |
| Ours w/ CRE (Qwen3.5-9B) | 79.5 | 64.2 | 37.9 |
| Ours w/ CRE (Qwen3.5-4B) | 79.3 | 64.3 | 37.5 |

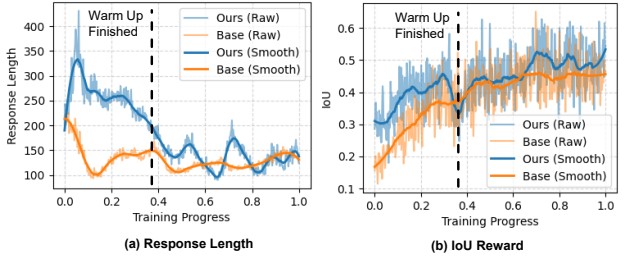

*Figure 3.* **Evolution of Response Length and IoU Reward.**

*Table 7.* **Ablation of Temporal-Sensitive Reward.**

| Setting | Charades-STA | | |
|---|---|---|---|
| | R1@0.3 | R1@0.5 | R1@0.7 |
| Shuffle Full Video | 78.1 | 63.5 | 36.2 |
| Random Cropping | 78.6 | 63.1 | 35.4 |
| Shuffle GT Boundary | **79.7** | **64.8** | **38.4** |

tribution of Constructive Reasoning Exploration (CRE), Temporal-Sensitive Reward (TR), and Progressive Curriculum (PC). As shown in Tab. 4, the baseline model (Row 1), which relies on standard RL, achieves 61.1% on R1@0.5. Adding the Temporal Reward alone (Row 2) brings a clear improvement to 63.1%, verifying that encouraging temporal-sensitive reasoning helps the model focus on valid cues. Using CRE alone without the subsequent self-exploration phase (Row 3 and Row 4) leads to a drastic performance drop. This suggests that simply imitating the constructed reasoning path is insufficient, as the constructed reasoning path cannot be obtained during testing and must rely on the model's own strategy for reasoning. Introducing CRE with the Progressive Curriculum (Row 5) improves performance to 63.9%, demonstrating the value of warm-up with high-quality reasoning traces and progressive curriculum learning. The best performance is achieved when all components are combined (Row 6), validating the necessity of our full framework.

**Analysis of Constructive Reasoning Exploration.** We first compare our approach with standard Supervised Fine-Tuning (SFT) using the constructed data. As shown in Tab. 5, SFT yields sub-optimal performance compared to our method. This may be because SFT relies on mimicking static demonstrations, while our method uses the dynamic reasoning variations through random sampling, enabling the model to learn to distinguish between informative visual cues and distractors. Regarding the construction strategy, Tab. 5 shows that randomly sampling captions works better than using all captions. Including all captions tends to introduce excessive noise, overwhelming the reasoning process. This is further supported by the evolution of response length in Fig. 3 (a). The average response length during the warmup phase reflects the average length of the reasoning paths we constructed; after warmup, the figure shows a de-

creasing trend. This indicates that the model learns to filter irrelevant information rather than exhaustively describing every event, ensuring high precision without compromising inference efficiency.

Then we analyze the loss used in the CRE phase. Tab. 5 shows that directly applying the on-policy GRPO loss to the off-policy constructed data leads to significant performance degradation. In contrast, our Advantage-Weighted Behavioral Cloning (AW-BC) loss effectively utilizes this data. Specifically, removing the weighting or positive advantage filter ($\mathbb{I}(A > 0)$) results in a performance drop. This demonstrates that the weighting and filter mechanisms are crucial: they allow the model to selectively imitate only the high-quality reasoning paths that yield higher reward.

We further analyze the training dynamics of the IoU reward in Fig. 3 (b). The CRE warm-up provides significantly higher initial rewards than the baseline, validating that the quality of constructed reasoning paths is better than random rollouts. Notably, we observe a temporary drop in IoU immediately after the warm-up finishes. This is expected, as the model transitions from imitating high-quality external reasoning traces to generating its own. As the self-exploration phase continues, the IoU steadily rises and eventually surpasses the peak performance of the CRE stage. This indicates that the model successfully discovers even better reasoning strategies through self-exploration, proving the necessity of the Progressive Curriculum strategy.

**Analysis of Different Dense Captioners.** To test the impact of lower-quality captions and potential noise propagation, we progressively downgraded the captioner from the powerful Gemini-3-Pro to the open-source Qwen3.5-9B, and further down to the much smaller Qwen3.5-4B. The results on Charades-STA are shown in Tab. 6. (1) Replacing Gemini with the open-source Qwen3.5-9B or Qwen3.5-4B yields

*Table 8.* **Training and inference time comparisons.**

| Method | Training Time (seconds per step) | | | | | Inference Speed (seconds per video) |
|---|---|---|---|---|---|---|
| | Rollout | $\pi_{ref}$ | $\pi_{\theta old}$ | $\pi_\theta$ | Total | |
| Baseline | 48.3 | 35.2 | 22.1 | 48.0 | 154.1 | 0.77 |
| **TaRO (Ours)** | 49.1 | 35.0 | 40.3 | 50.6 | 175.3 | 0.78 |

*Table 9.* **Performance of disabling reasoning at inference.**

| Setting | Charades-STA | | ActivityNet | | QVHighlights | |
|---|---|---|---|---|---|---|
| | R1@0.3 | R1@0.5 | R1@0.3 | R1@0.5 | R1@0.3 | R1@0.5 |
| w/ reasoning at inference | 79.7 | 64.8 | 60.6 | 39.8 | 82.6 | 69.4 |
| w/o reasoning at inference | 79.5 | 64.8 | 59.1 | 38.9 | 82.3 | 67.9 |

only marginal performance degradation. This indicates that our model is relatively robust to dense captions of varying quality. This is because our Constructive Reasoning primarily serves to teach the model the pattern of thinking with time and how to selectively attend to crucial visual cues, rather than requiring flawless, oracle-level descriptions. (2) Even if we completely remove the external captioner (Ours w/o CRE) to eliminate any possibility of cascading errors, introducing only our Temporal Reward still significantly outperforms the second-best method, Time-R1.

**Analysis of Temporal Reward.** In Tab. 7, we examine the design choices for the Temporal-Sensitivity Reward. Regarding the perturbation method, "Shuffle GT Boundary" outperforms global shuffling ("Shuffle Full Video") and random cropping. Shuffling only the frames near the ground-truth boundaries provides a stricter test of whether the reasoning is precisely anchored to the start and end events, whereas global shuffling is too easy for the model to distinguish. We include the hyper-parameter analysis $(\Delta t, \tau, \alpha)$ in the Appendix A.

**Analysis of Training and Inference Efficiency.** During training, our method requires an additional forward pass for shuffled videos, which increases the per-step training time by only 13.8% as shown in Tab. 8. This is because GRPO also requires multiple forward passes to generate rollouts and compute $\pi_{ref}$, $\pi_{\theta old}$, and $\pi_\theta$. Our method only adds one extra forward pass for $\pi_{\theta old}$. The inference speed is comparable with the baseline, as the final average response length is similar to the baseline, as shown in Fig. 3 (a).

**Analysis of Disabling Reasoning at Inference.** As shown in Tab. 9, the impact of reasoning during inference depends on query complexity: For simple queries (e.g., Charades-STA), performance remains stable without reasoning, indicating TaRO internalizes key cue selection and can output answers directly for simpler events, saving tokens while maintaining accuracy. For complex queries (e.g., ActivityNet and QVHighligts), disabling inference reasoning reduces performance, showing intermediate reasoning traces are crucial to disentangle complex events.

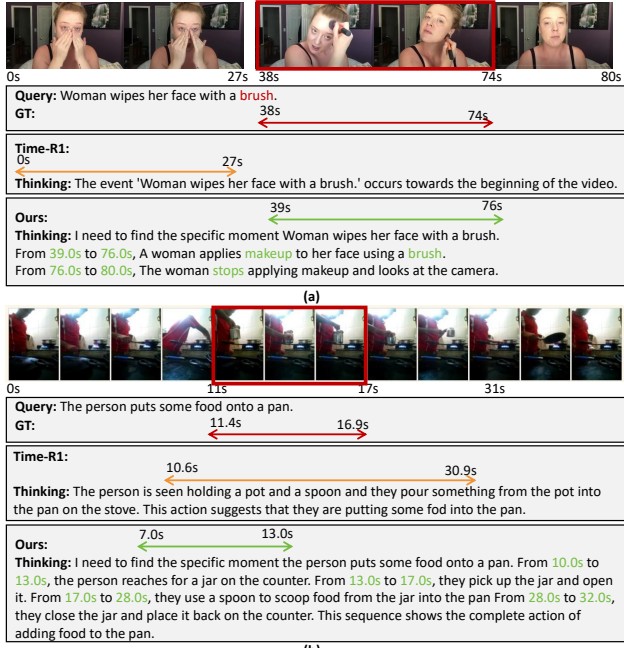

*Figure 4.* **Qualitative results of our TaRO and Time-R1.**

### 4.5. Qualitative Results

Fig. 4 compares TaRO with the Time-R1. In Fig. 4 (a), the video starts with a distractor (wiping face with hands) that visually resembles the query (with a brush). The baseline is misled, producing vague reasoning and incorrect localization. In contrast, TaRO generates precise reasoning anchored to specific timestamps and visual details (identifying the "brush"). This demonstrates our model's ability to explicitly think with time through temporal-aware reasoning, leading to accurate grounding. We also observed that the model sometimes shows inconsistency between its intermediate reasoning and final prediction. In Fig. 4 (b), the model's reasoning process can identify key actions with precise timestamps. However, it occasionally fails to leverage its own reasoning, producing a final temporal segment that contradicts the timestamps identified during reasoning. More results are shown in Appendix C.

### 5. Conclusion

In this paper, we propose TaRO, a novel framework to enhance temporal reasoning in video temporal grounding. We propose a Constructive Reasoning Exploration strategy to facilitate the efficient discovery of high-quality reasoning paths, and a Temporal-Sensitivity Reward to evaluate the quality of reasoning and encourage reasoning anchored in visual-temporal evidence. Extensive experiments show that TaRO significantly outperforms existing methods, achieving state-of-the-art performance on VTG benchmarks.

## Impact Statement

This paper presents work whose goal is to enhance the temporal reasoning capabilities of Multi-modal Large Language Models (MLLMs) in video temporal grounding. By encouraging models to think with time, our framework aims to provide more interpretable and precise localization of events in untrimmed videos. This advancement has potential societal benefits in improving video accessibility, automated content retrieval, and intelligent surveillance analysis. The primary impact of this work is to advance the quality of visual-temporal reasoning in machine learning.

## Acknowledgements

This work was supported by the grants from the National Natural Science Foundation of China (62372014, 62525201, 62132001, 62432001), Beijing Nova Program and Beijing Natural Science Foundation (4252040, L247006).

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

## A. Analysis of Hyper-parameters

*Table 10.* **Ablation of Hyper-parameters of Temporal-Sensitive Reward.**

| Setting | Charades-STA | | | |
|---|---|---|---|---|
| | R1@0.3 | R1@0.5 | R1@0.7 | mIoU |
| $\Delta t = 2$ | **79.8** | 64.1 | 37.5 | 55.2 |
| $\Delta t = 3$ | 79.7 | **64.8** | **38.4** | **55.5** |
| $\Delta t = 4$ | 78.6 | 64.6 | 37.8 | 54.9 |
| $\tau = 0.3$ | 78.0 | 61.8 | 35.9 | 53.7 |
| $\tau = 0.5$ | 79.5 | 63.6 | 37.8 | 55.1 |
| $\tau = 0.7$ | **79.7** | **64.8** | **38.4** | **55.5** |
| $\tau = 1.0$ | 78.9 | 63.9 | 36.3 | 54.4 |
| $\alpha = 0.2$ | 77.6 | 62.4 | 35.8 | 53.7 |
| $\alpha = 0.3$ | **79.7** | 64.8 | **38.4** | **55.5** |
| $\alpha = 0.4$ | 79.5 | **65.4** | 36.9 | 54.9 |

In this section, we analyze the impact of hyper-parameters in the Temporal-Sensitive Reward: the shuffle window size $\Delta t$, the IoU threshold $\tau$, and the reward weight $\alpha$. The results are summarized in Table 10.

For the Shuffle Window Size ($\Delta t$), we observe that the performance is relatively stable for the shuffle window size within a reasonable range. $\Delta t = 3$s yields the optimal results across most metrics.

For the IoU Threshold ($\tau$), the results show that $\tau = 0.7$ achieves the best performance. This confirms the importance of filtering: the temporal sensitivity reward is most effective when applied only to reasonably accurate predictions (IoU $\geq$ 0.7). Lower thresholds (e.g., $\tau = 0.3$) result in a noticeable performance drop, as they may encourage sensitivity even for incorrect or low-quality predictions. When $\tau = 1.0$, which effectively removes the temporal-sensitive reward, model performance also drops noticeably, confirming the effectiveness of the temporal-sensitive reward.

Finally, for the reward weight $\alpha$, we find that a moderate value of 0.3 strikes the best balance between encouraging temporal sensitivity and optimizing the primary grounding objective. Setting $\alpha = 0.3$ achieves the highest R1@0.3 and R1@0.7, whereas lower or higher weights lead to suboptimal trade-offs.

## B. Implementation Details

Our training follows our proposed progressive curriculum, consisting of 1 epoch for the warm-up phase (with constructive reasoning), followed by 2 epochs of the self-exploration phase (with random rollout). Regarding the hyper-parameters, we set $G = 8$, $\Delta t = 3$, $\alpha = 0.3$, $\tau = 0.7$. We perform full-parameter fine-tuning of the language model parameters with a learning rate of $1 \times 10^{-6}$, while the vision encoder remains frozen throughout all stages. For video input processing, uniformly sample frames at a rate of 2 FPS and constrain the total number of video tokens to be approximately 3584. All the experiments are conducted on 8 HUAWEI Ascend 910B NPUs.

The prompt used for dense captions generation in our Constructive Reasoning Exploration is shown in Fig. 5. To evaluate the video temporal grounding performance, we employ specific prompts tailored to different model configurations, as illustrated in Fig. 5. For our model, we utilize the "VTG Prompts for Our Model". For the Qwen2.5-VL (Bai et al., 2025b) and Qwen3-VL (Bai et al., 2025a) baseline, we use the "VTG Prompt for Non-Reasoning Models" to obtain direct timestamp predictions. For other comparative methods, such as Time-R1 (Wang et al., 2026) and UniTime (Li et al., 2026b), we adhere to the official prompts provided in their respective papers.

## C. Additional Qualitative Results

Fig. 6 and Fig. 7 present further qualitative comparisons between our proposed TaRO and the baseline Time-R1. As shown in Fig. 6 (a), for the query "a woman in red leggings attempts to climb a rope ladder", Time-R1 provides a superficial reasoning trace that merely restates the query and hallucinates a time range (10s-25s), leading to poor alignment with the ground truth (22s-40s). In contrast, TaRO demonstrates a granular understanding of the video content. Its reasoning trace

**Prompts for Dense Captions**

To accurately pinpoint the event "{query}" in the video, determine the precise time period of the event.

Please output your response in JSON format with the following keys:
- "dense_captions": A list of strings, each describing a key event and its time period using **MM:SS format** (e.g., "From 00:03 to 00:09, [description]"). These captions must be **chronological**, **dense**, and **specifically focused on distinguishing actions** relevant to the query to assist in temporal grounding.
- "answer": A list containing two strings in **MM:SS format** (e.g., ["00:08", "00:12"]) representing the predicted time range for the query.

Example of the expected output format and content depth:
Query: "eating from a box"
{{
  "dense_captions": [
    "From 00:00 to 00:05, a person is holding a box of food and talking to the camera, but they are not eating yet.",
    "From 00:05 to 00:08, the person opens the box and looks inside. Still no eating action.",
    "At 00:08, the person picks up a piece of food from the box.",
    "From 00:08 to 00:12, the person puts the food in their mouth and chews while holding the box. This clearly matches 'eating from a box'.",
    "From 00:12 to 00:15, the person puts the box down and wipes their mouth. The action has ended."
  ],
  "answer": ["00:08", "00:12"]
}}

**VTG Prompt for Non-Reasoning Models**

You are given a video. Please find the visual event described by a sentence in the video, determining its starting and ending timestamps. The format should be: 'The event happens in the start time - end time seconds'. Now I will give you the textual sentence: {query}. Please return its start time and end timestamps.

**VTG Prompts for Our Model**

To accurately pinpoint the event "{query}" in the video, determine the precise time period of the event.

Output your thought process within the <think> </think> tags, including analysis with either specific time ranges (xx.xx to xx.xx) in <timestep> </timestep> tags.

Then, provide the start and end times (in seconds, precise to two decimal places) in the format "start time to end time" within the <answer> </answer> tags.

*Figure 5.* The prompt used for dense caption generation and model evaluation.

explicitly identifies the target action starting at 19.0s and "she falls onto a yellow mat" at 37.0s. By actively *thinking with time*, TaRO successfully predicts a more precise interval (19s-37s). As shown in Fig. 7 case 1, the query is "Watch the heat". Our method successfully associates this abstract phrase with the visual cue of a microwave and accurately grounds it to the specific event. In Fig. 7 case 2, the query is "where did I throw the dust". The Time-R1 baseline exhibits a strong logical bias in its reasoning, stating: "The event happens after the person has finished cleaning the kitchen and is likely disposing of the dust or trash. This would be towards the end of the video." Consequently, it fails because the event actually occurs in the middle of the video. In contrast, our method avoids biased assumptions and strictly relies on visual evidence, successfully grounding the key cues and precise timestamps: "From 78.0s to 81.0s, the person walks towards a trash can located near the doorway. From 81.0s to 82.0s, the person throws the dust into the trash can."

We also analyze a failure case in Fig. 6 (b) and Fig. 7. In Fig. 6 (b), the query "Three men are standing on a mat" corresponds to a ground truth (7s-69s) that broadly encompasses the entire scene, including segments where the actors are speaking or performing actions. Time-R1 predicts a shorter segment (16s-29s) with generic reasoning. However, TaRO predicts a strictly grounded interval (19s-30s), explicitly reasoning that the men are specifically "standing on a mat" only during this period. It further notes that from 30.0s to 38.0s, the man is "speaking", and from 38.0s to 69.0s, he is "performing capoeira moves". While this strict adherence to visual evidence results in a mismatch with the loose ground truth annotation and a lower IoU, it demonstrates that TaRO has learned to ground actions based on rigorous visual-temporal cues. In Fig. 6 (b) failure case 1, due to the inherent ambiguity and subjectivity of temporal boundaries in untrimmed videos, the model's prediction preferences sometimes fail to perfectly align with human annotation standards. This misalignment can result in predicted segments that are either overly long or fragmented. In Fig. 6 (b) failure case 2, we found that the model is relatively insensitive to the precise endpoints of fast, dynamic actions. For example, when queried with "person put it back closed the shelf", the model's predicted segment incorrectly includes a significant portion of the static scene after the shelf has already been closed.

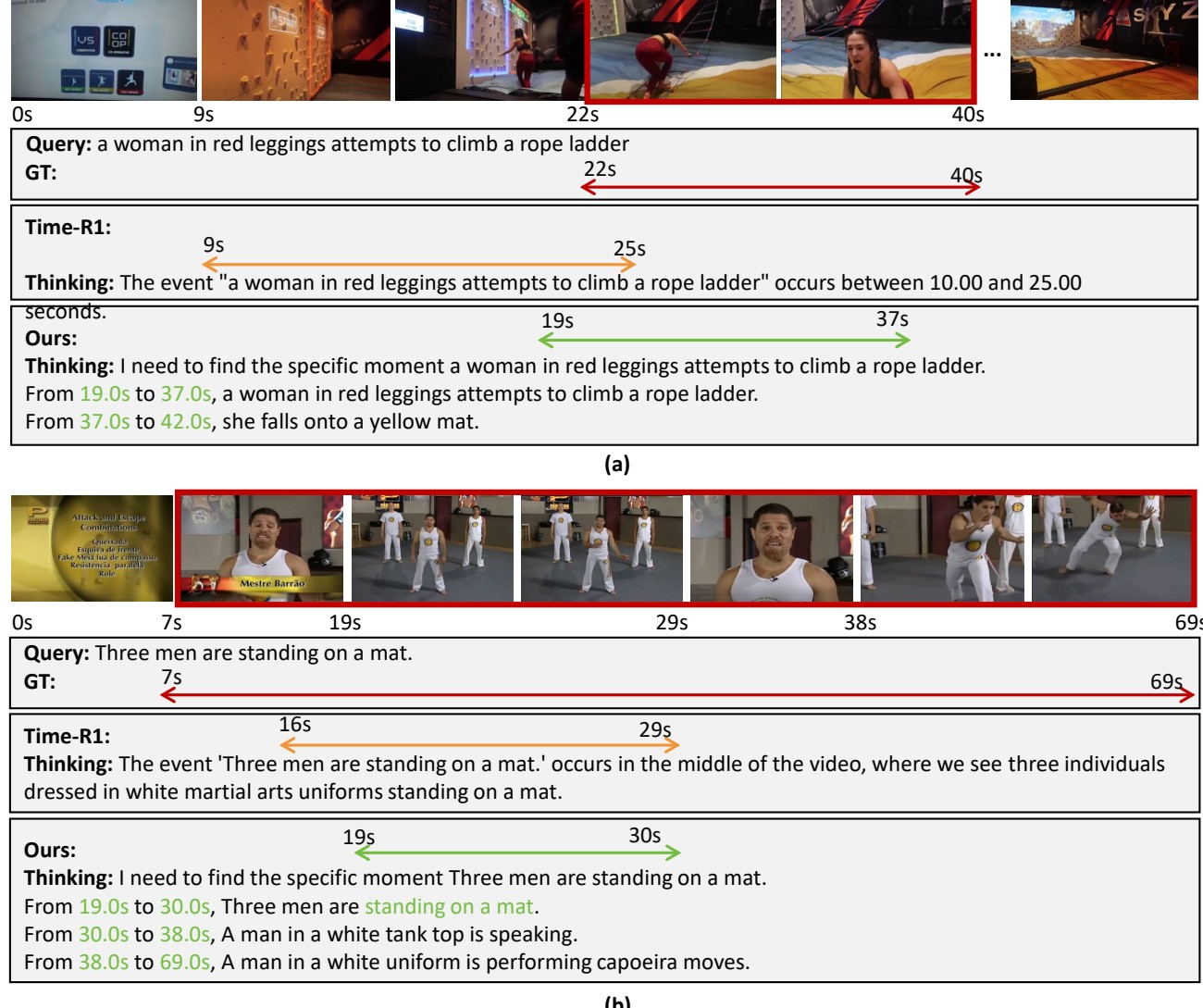

*Figure 6.* **More Qualitative results of our TaRO and Time-R1.**

**Case 1:**

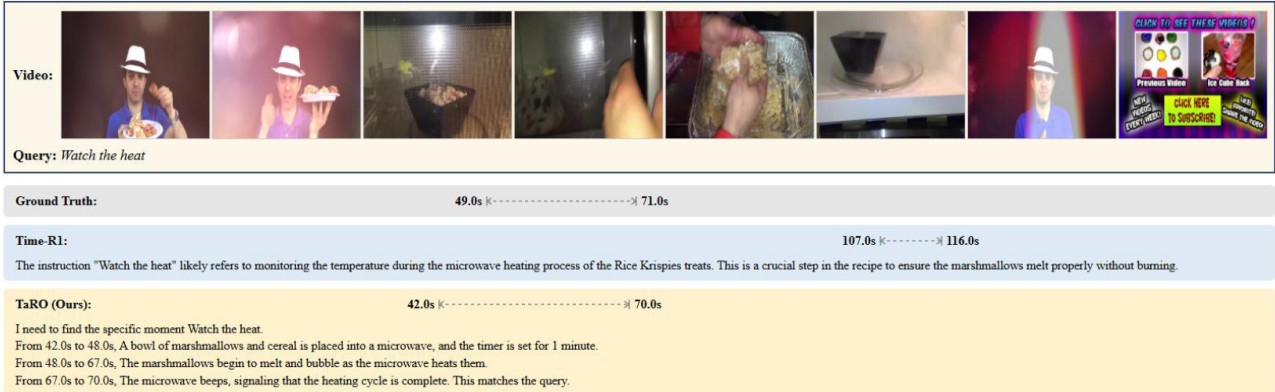

**Video:**

**Query:** *Watch the heat*

**Ground Truth:**         49.0s |◁- - - - - - - - - - - - - - - - - ▷| 71.0s

**Time-R1:**         107.0s |◁- - - - - - ▷| 116.0s

The instruction "Watch the heat" likely refers to monitoring the temperature during the microwave heating process of the Rice Krispies treats. This is a crucial step in the recipe to ensure the marshmallows melt properly without burning.

**TaRO (Ours):**         42.0s |◁- - - - - - - - - - - - - - - - ▷| 70.0s

I need to find the specific moment Watch the heat.
From 42.0s to 48.0s, A bowl of marshmallows and cereal is placed into a microwave, and the timer is set for 1 minute.
From 48.0s to 67.0s, The marshmallows begin to melt and bubble as the microwave heats them.
From 67.0s to 70.0s, The microwave beeps, signaling that the heating cycle is complete. This matches the query.

**Case 2:**

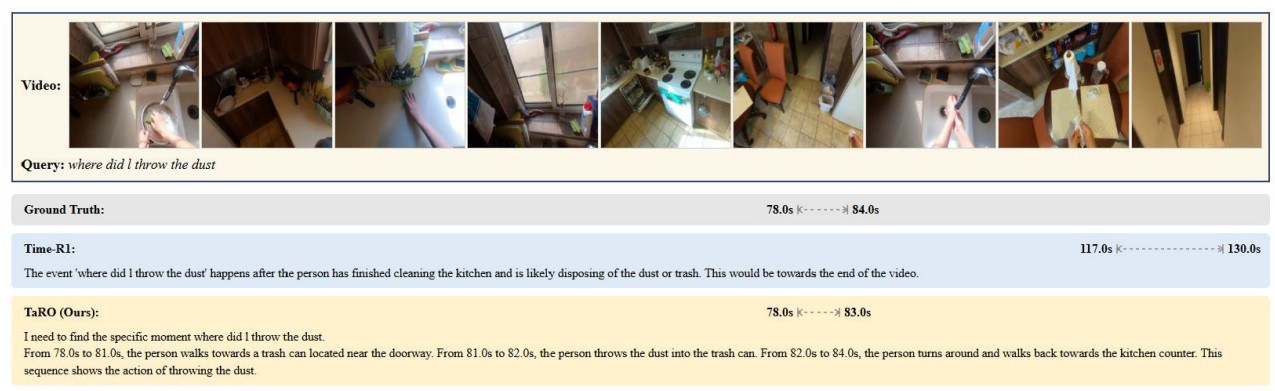

**Video:**

**Query:** *where did l throw the dust*

**Ground Truth:**         78.0s |◁- - - - - ▷| 84.0s

**Time-R1:**         117.0s |◁- - - - - - - - - - - - - - ▷| 130.0s

The event 'where did l throw the dust' happens after the person has finished cleaning the kitchen and is likely disposing of the dust or trash. This would be towards the end of the video.

**TaRO (Ours):**         78.0s |◁- - - - - ▷| 83.0s

I need to find the specific moment where did l throw the dust.
From 78.0s to 81.0s, the person walks towards a trash can located near the doorway. From 81.0s to 82.0s, the person throws the dust into the trash can. From 82.0s to 84.0s, the person turns around and walks back towards the kitchen counter. This sequence shows the action of throwing the dust.

**Failure Case 1:**

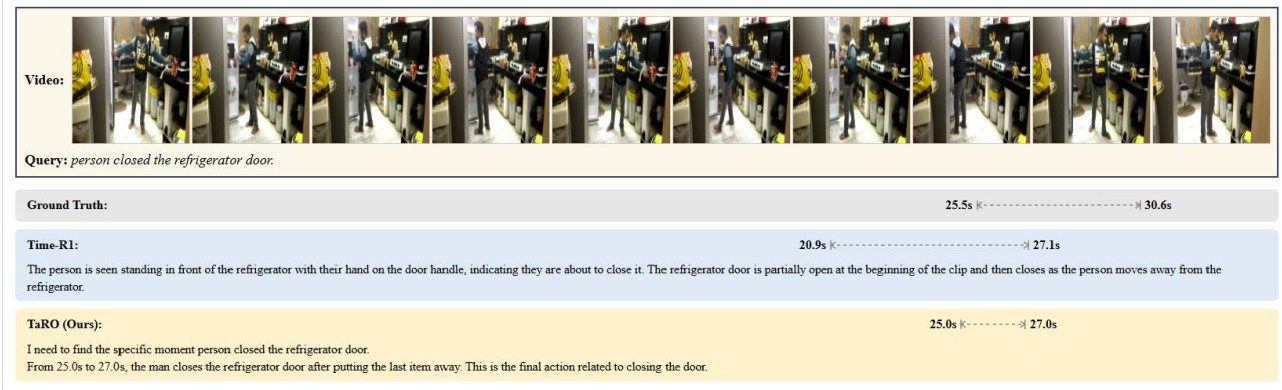

**Video:**

**Query:** *person closed the refrigerator door.*

**Ground Truth:**         25.5s |◁- - - - - - - - - - - - - - - - - ▷| 30.6s

**Time-R1:**         20.9s |◁- - - - - - - - - - - - - - - - - - - ▷| 27.1s

The person is seen standing in front of the refrigerator with their hand on the door handle, indicating they are about to close it. The refrigerator door is partially open at the beginning of the clip and then closes as the person moves away from the refrigerator.

**TaRO (Ours):**         25.0s |◁- - - - - - - ▷| 27.0s

I need to find the specific moment person closed the refrigerator door.
From 25.0s to 27.0s, the man closes the refrigerator door after putting the last item away. This is the final action related to closing the door.

**Failure Case 2:**

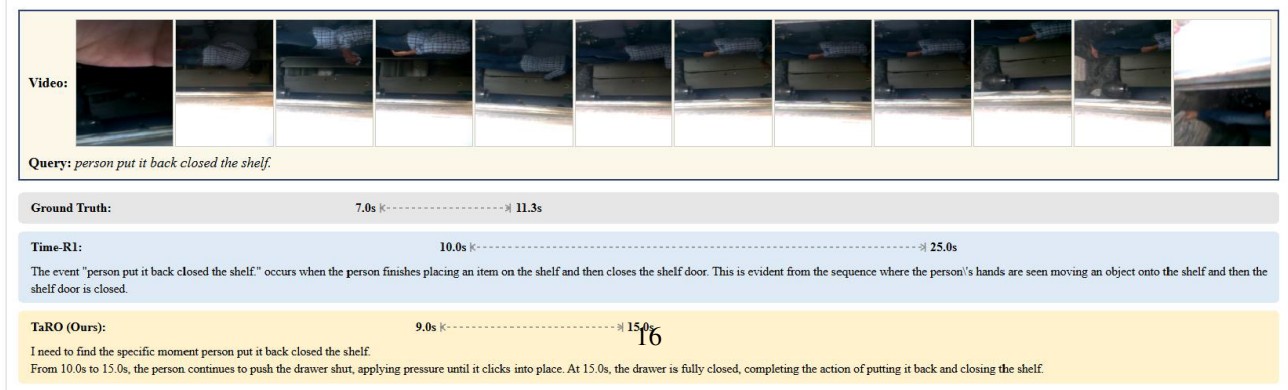

**Video:**

**Query:** *person put it back closed the shelf.*

**Ground Truth:**         7.0s |◁- - - - - - - - - - - - - - ▷| 11.3s

**Time-R1:**         10.0s |◁- - - - - - - - - - - - - - - - - - - - - - - - ▷| 25.0s

The event "person put it back closed the shelf." occurs when the person finishes placing an item on the shelf and then closes the shelf door. This is evident from the sequence where the person\'s hands are seen moving an object onto the shelf and then the shelf door is closed.

**TaRO (Ours):**         9.0s |◁- - - - - - - - - - - - - - - - - - - ▷| 15.0s

I need to find the specific moment person put it back closed the shelf.
From 10.0s to 15.0s, the person continues to push the drawer shut, applying pressure until it clicks into place. At 15.0s, the drawer is fully closed, completing the action of putting it back and closing the shelf.

*Figure 7.* **More Qualitative results and failure cases of our TaRO and Time-R1.**