# OpenReview forum: "Temporal-Aware Reasoning Optimization for Video Temporal Grounding"
_ICML.cc/2026/Conference — ICML 2026 regular_

### Official Review · Reviewer_qvR8 · 2026-03-12

**Soundness:** 3
**Presentation:** 3
**Significance:** 3
**Originality:** 3
**Overall Recommendation:** 4
**Confidence:** 4

**Summary:**

This paper tackles the video temporal grounding problem, specifically addressing the "superficial reasoning" trap often found in current RL approaches. The authors argue that existing RL methods suffer from poor exploration and misaligned rewards, leading to reasoning chains that don't actually help the model predict timestamps.

To solve this, they introduce TaRO (Temporal-Aware Reasoning Optimization). It’s a two-stage framework: first, it uses Constructed Reasoning Traces (CRE) to "warm up" the model’s temporal logic; then, it transits to RL rollouts where a  specialized temporal reward guides the model’s self-generated reasoning. The evaluation convers multiple base models and benchmarks, showing consistent performance gains.

**Compliance With Llm Reviewing Policy:**

Affirmed.

**Key Questions For Authors:**

- It’s unclear how robust TaRO is to the quality of the CRE. What happens if the captions are noisy? Some ablation or discussion on the caption quality would be helpful.

**Limitations:**

I’d like to see a more candid discussion of the failure cases.

**Strengths And Weaknesses:**

### Strengths
- The core observation: reasoning chains in current models often fail to actually improve grounding, is a timely point for the field.

 - The "temporal-sensitivity reward" is a simple and effective idea. It directly addresses the problem by measuring how well the reasoning actually anchors to the video.

- The results are convincing, achieving SOTA across several benchmarks and demonstrating that the method scales well across different model sizes.

### Weaknesses
- The approach depends heavily on external dense captioning. I’m concerned about the cascading error effect. if these captions are low-quality or contain hallucinations, does that noise propagate through the reasoning stage and ultimately hurt grounding performance?


- While the numbers are good, the qualitative side feels a bit thin. With only three examples between the main paper and the supp, it’s hard to get a broad sense of where the model truly shines. More diverse visual examples would strengthen the case.

---

> ### Author Rebuttal · Authors · 2026-03-31
>
> # Q1: The approach depends heavily on external dense captioning. If these captions are low-quality or contain hallucinations, does that noise propagate through the reasoning stage and ultimately hurt grounding performance?
>
> To test the impact of lower-quality captions and potential noise propagation, we progressively downgraded the captioner from the powerful Gemini-3-Pro to the open-source Qwen3.5-9B, and further down to the much smaller Qwen3.5-4B. The results on Charades-STA are shown below.
>
> (1) Replacing Gemini with the open-source Qwen3.5-9B or Qwen3.5-4B yields only marginal performance degradation. This indicates that our model is relatively robust to dense captions of varying quality. This is because our Constructive Reasoning primarily serves to teach the model the pattern of thinking with time and how to selectively attend to crucial visual cues, rather than requiring flawless, oracle-level descriptions.
>
> (2) As shown in the last row, even if we completely remove the external captioner to eliminate any possibility of cascading errors, introducing only our Temporal Reward still significantly outperforms the second-best method, Time-R1.
>
> | **Method** | **R1@0.3** | **R1@0.5** | **R1@0.7** |
> | --- | --- | --- | --- |
> | Time-R1 | 78.1 | 60.8 | 35.3 |
> | Ours w/ Constructive Reasoning (Gemini-3-Pro) | **79.7** | **64.8** | **38.4** |
> | Ours w/ Constructive Reasoning (Qwen3.5-9B) | 79.5 | 64.2 | 37.9 |
> | Ours w/ Constructive Reasoning (Qwen3.5-4B) | 79.3 | 64.3 | 37.5 |
> | Ours w/o Constructive Reasoning | 78.6 | 63.1 | 36.1 |
>
> (3) We found that Gemini's grounding performance is actually not very good. This confirms that our gains do not come from simply distilling a superior model's grounding ability. Gemini only provides raw, unselected atomic events, while our framework explicitly teaches the model the crucial skill of thinking with time and selecting relevant visual cues.
>
> | **Method** | **R1@0.3** | **R1@0.5** | **R1@0.7** |
> | --- | --- | --- | --- |
> | Gemini | 39.1  | 24.4  | 12.8 |
> | Time-R1 | 41.8  | 29.4  | 16.4 |
> | Ours | 54.6  | 37.8  | 20.0 |
>
> # Q2: The qualitative side feels a bit thin.
>
> We provide more visualizations to compare our model with the baseline, Time-R1 at [https://anonymous.4open.science/api/repo/taro_vis-23FE/file/vis.png](https://anonymous.4open.science/api/repo/taro_vis-23FE/file/vis.png?v=b9144f56), including both successful and failure cases.
>
> As we can see, in case 1, the query is *"Watch the heat"*. Our method successfully associates this abstract phrase with the visual cue of a microwave and accurately grounds it to the specific event. In case 2, the query is *"where did I throw the dust"*. The Time-R1 baseline exhibits a strong logical bias in its reasoning, stating: *"The event happens after the person has finished cleaning the kitchen and is likely disposing of the dust or trash. This would be towards the end of the video."* Consequently, it fails because the event actually occurs in the middle of the video. In contrast, our method avoids biased assumptions and strictly relies on visual evidence, successfully grounding the key cues and precise timestamps: *"From 78.0s to 81.0s, the person walks towards a trash can located near the doorway. From 81.0s to 82.0s, the person throws the dust into the trash can."*
>
>  Discussion of the failure cases can be found in our response to **Q3**.
>
> # Q3: I’d like to see a more candid discussion of the failure cases.
>
> We provide visualizations of failure cases at [https://anonymous.4open.science/api/repo/taro_vis-23FE/file/vis.png](https://anonymous.4open.science/api/repo/taro_vis-23FE/file/vis.png?v=b9144f56). Based on our observations, our failure cases primarily fall into the following categories.
>
> (1) Misalignment of Event Boundaries: Due to the inherent ambiguity and subjectivity of temporal boundaries in untrimmed videos, the model's prediction preferences sometimes fail to perfectly align with human annotation standards. This misalignment can result in predicted segments that are either overly long or fragmented, as illustrated in the visualized Failure Case 1.
>
> (2) Insensitivity to Highly Dynamic Actions: We found that the model is relatively insensitive to the precise endpoints of fast, dynamic actions. For example, when queried with *"person put it back closed the shelf"* (Failure Case 2), the model's predicted segment incorrectly includes a significant portion of the static scene *after* the shelf has already been closed.
>
> (3) Reasoning-Prediction Inconsistency: We observed that the model sometimes shows inconsistency between its intermediate reasoning and final prediction. As illustrated in Failure Case 3, the model’s reasoning process can identify key actions with precise timestamps. However, it occasionally fails to leverage its own reasoning, producing a final temporal segment that contradicts the timestamps identified during reasoning.

---

> > ### Author Rebuttal · Reviewer_qvR8 · 2026-04-04
> >
> > Thanks the authors for the response.

---

> > > ### Author Response · Authors · 2026-04-07
> > >
> > > Dear Reviewer qvR8,
> > >
> > > Thank you for your thoughtful feedback. We appreciate your acknowledgment of our focus on superficial reasoning and the effectiveness of our temporal-sensitivity reward. We are pleased to have addressed all your concerns. In the final version, we will include ablation studies on different captioner qualities, expanded qualitative visualizations, and a detailed discussion of our failure cases.
> > >
> > > Regards,
> > >
> > > Authors

---

### Official Review · Reviewer_Buer · 2026-03-12

**Soundness:** 3
**Presentation:** 3
**Significance:** 2
**Originality:** 2
**Overall Recommendation:** 4
**Confidence:** 3

**Summary:**

This paper proposes TaRO, a reinforcement learning framework designed to improve the quality of temporal reasoning in multimodal large language models for Video Temporal Grounding (VTG). The method utilizes Constructive Reasoning Exploration (CRE) to warm-start the policy with timestamped captions, a Temporal-Sensitivity Reward to penalize superficial reasoning by shuffling frames near ground-truth boundaries, and a progressive curriculum to transition to on-policy exploration.

**Compliance With Llm Reviewing Policy:**

Affirmed.

**Final Justification:**

The authors's rebuttal have resolved my main concerns and I have decided to raise my score for this submission.

**Key Questions For Authors:**

1. In Figure 5, the prompt for the dense captioner explicitly requests an "answer" interval. Is this "answer" field ever used during training, filtering, or rollout construction? Can you provide an ablation using an open-source captioner or query-agnostic captions?
2. What is the training-time overhead introduced by generating dense captions and computing the Temporal-Sensitivity Reward, which requires two likelihood passes per rollout, compared to the Time-R1 baseline?

**Limitations:**

I recommend adding a section that explicitly acknowledges the computational overhead caused by generating dense captions and performing dual likelihood evaluations for the reward. Additionally, the authors should discuss the framework's reliance on proprietary models like Gemini-3-Pro.

**Strengths And Weaknesses:**

strengths:
1. The paper addresses a clear failure mode in current RL-based VTG models: the generation of superficial video narrations that do not functionally assist in temporal localization.
2. The training pipeline is logically designed. Transitioning from off-policy imitation learning via Advantage-Weighted Behavioral Cloning (AW-BC) to on-policy exploration is a sound approach to handle large reasoning spaces.
3. The method shows consistent empirical improvements across four VTG benchmarks compared to strong baselines like Time-R1.

weaknesses:
1. CRE relies on a proprietary model (Gemini-3-Pro) prompted to explicitly output answer intervals. This provides query-conditioned pseudo-grounding supervision, confounding whether the gains come from the proposed RL exploration or the external teacher model. This also raises reproducibility concerns.
2. The claim that TaRO improves reasoning quality lacks causal proof. The paper does not provide an ablation comparing TaRO with and without reasoning at inference to show that the generated reasoning, rather than the training process alone, drives the performance gains.
3. The Temporal-Sensitivity Reward is unvalidated. It is unclear if the log-probability drop correlates with better semantic reasoning or simply captures model sensitivity to low-level pixel disturbances near event boundaries.
4. The computational overhead is unquantified. The paper must report the impact of generating dense captions and running dual likelihood evaluations on training time and overall efficiency compared to baselines.
5. The data-efficiency claims lack a proper comparative baseline. TaRO should be compared against Time-R1 under the exact same reduced-data constraints, rather than comparing against different methods trained on the full dataset.

---

> ### Author Rebuttal · Authors · 2026-03-31
>
> # Q1: Ablation with open-source captioner.
>
> See **Reviewer qvR8, Q1** for ablation on open-source captioners and without external teachers.
>
> # Q2: Ablation with query-agnostic captions.
>
> We compared Qwen3.5-9B generating query-relevant vs. query-agnostic captions on Charades-STA. Removing query guidance degrades performance more than switching to a smaller captioner, as videos contain rich, multi-perspective content, and query-agnostic captions often miss key clues, limiting our CRE stage from constructing high-quality reasoning paths. Even so, our method still outperforms Time-R1.
>
> | **Method** | **R1@0.3** | **R1@0.5** | **R1@0.7** |
> | --- | --- | --- | --- |
> | Time-R1 | 78.1 | 60.8 | 35.3 |
> | Ours w/  Gemini | 79.7 | 64.8 | 38.4 |
> | Ours w/  Qwen3.5-9B | 79.5 | 64.2 | 37.9 |
> | Ours w/  Qwen3.5-9B, query-agnostic | 79.0 | 63.8 | 37.3 |
>
> # Q3: The prompt requests an answer interval. Is this used anywhere?
>
> The answer field is not used in training or inference; it is requested only to encourage query-relevant captions. Because we found that Gemini's grounding performance is actually not very good (See **Reviewer RHKL, Q2 (3)**).
>
> # Q4: TaRO with and without reasoning at inference.
>
> (1) As shown in Fig. 1 of the main text, unlike the baseline, where training with reasoning has little contribution to the final grounding performance, explicitly **training** our model to reason yields significant gains. This proves our training paradigm effective.
>
> (2) The impact of reasoning during **inference** depends on query complexity: For simple queries (Charades), performance remains stable without reasoning, indicating TaRO internalizes key cue selection and can output answers directly for simpler events, saving tokens while maintaining accuracy. For complex queries (ActivityNet & QVHighligts), disabling inference reasoning reduces performance, showing intermediate reasoning traces are crucial to disentangle complex events.
>
> | **Method** | **Charades** |  | **ActivityNet** |  | **QVHighlights** |  |
> | --- | --- | --- | --- | --- | --- | --- |
> |  | **R1@0.3** | **R1@0.5** | **R1@0.3** | **R1@0.5** | **R1@0.3** | **R1@0.5** |
> | Time-R1 | 78.1 | 60.8 | 58.6 | 39.0 | 80.3 | 66.2 |
> | Ours | 79.7 | 64.8 | 60.6 | 39.8 | 82.6 | 69.4 |
> | Ours Disable Inference Reasoning | 79.5 | 64.8 | 59.1 | 38.9 | 82.3 | 67.9 |
>
> # Q5: The log-probability drop correlates with better reasoning or simply captures model sensitivity to disturbances near event boundaries.
>
> Ablation in Tab. 3 of the main text shows that introducing only the Temporal Reward improves R1@0.5 from 61.1% to 63.1% on Charades-STA. This indicates the log-probability drop reflects semantic reasoning, not just sensitivity to boundary disturbances.
>
> More analysis is in **Reviewer AiHA, Q4.**
>
> # Q6: Impact of dense captions and dual likelihood evaluations on training time and efficiency.
>
> **(1) Training time and impact of dual likelihood.** See **Reviewer AiHA, Q2.**
>
> **(2) Impact of dense captions.** Preparing dense captions is a one-time offline preprocessing, taking approximately 6h. During training, rollout generation time also matches the baseline because our CRE teaches the model to only selectively describe key visual cues (Time comparisons are in **Reviewer AiHA, Q2**).
>
> **(3) Inference efficiency.** Inference speed on Charades-STA is also comparable to Time-R1 and only ~7% slower than a direct-prediction model without reasoning.
>
> | **Method** | **Speed (second per video)** | **R1@0.3 on Charades** |
> | --- | --- | --- |
> | Time-R1 | 0.77 | 78.1 |
> | Ours | 0.78 | 79.7 |
> | Ours w/o Reasoning | 0.73 | 79.5 |
>
> # Q7: Data-efficiency claims lack a proper comparative baseline.
>
> We fine-tuned Time-R1 with the same ActivityNet data ratios. Across all ratios, our method consistently outperforms Time-R1, confirming the think with time paradigm adapts more efficiently to downstream distributions.
>
> | Data Ratio | **Method** | **R1@0.3** | **R1@0.5** | **R1@0.7** |
> | --- | --- | --- | --- | --- |
> | 10% | Time-R1 |  70.9 | 53.4 | 31.8 |
> |  | Ours |  72.0  | 54.2  | 32.5 |
> | 30% | Time-R1 | 71.6 | 54.1 | 31.1 |
> |  | Ours | 73.0  | 55.3  | 33.2 |
> | 50% | Time-R1 | 72.2 | 56.1 | 32.9 |
> |  | Ours | 74.6  | 57.2  | 34.8 |

---

> > ### Author Rebuttal · Reviewer_Buer · 2026-04-03
> >
> > Thank you for your rebuttal and the clarifications provided. My main concerns have been resolved. Therefore, I have decided to raise my score for this submission.

---

> > > ### Author Response · Authors · 2026-04-07
> > >
> > > Dear Reviewer Buer,
> > >
> > > Thanks for increasing the score, and we appreciate it. We are encouraged by your recognition of our identification of shallow reasoning in VTG models, our training pipeline, and the consistent improvements across benchmarks. Following your recommendation, we will include clarifications on the impact of different captioners, inference reasoning, computational overhead, and data-efficiency comparisons in our final version.
> > >
> > > Regards,
> > >
> > > Authors

---

### Official Review · Reviewer_RHKL · 2026-03-13

**Soundness:** 3
**Presentation:** 3
**Significance:** 3
**Originality:** 3
**Overall Recommendation:** 5
**Confidence:** 3

**Summary:**

This paper focuses on Visual Temporal Grounding (VTG) tasks, where models need to localize the precise temporal segmentations within an untrimmed video. Authors claim that existing RL methods for VTG, such as Time-R1, often explore shallow reasoning, because they rely on random rollout, and current reward design mainly uses IoU (final result), without the access to the reasoning path. To address this, they propose TaRO (Temporal-aware Reasoning Optimization), which integrates (1) Constructive Reasoning Exploration (CRE) for building useful reasoning traces from visual cues and (2) time-sensitive rewards enforcing reasoning quality and adherence to the task. Extensive experiments have demonstrated that TaRO achieves a favorable performance.

**Compliance With Llm Reviewing Policy:**

Affirmed.

**Final Justification:**

The author's response addresses my concerns. I've decided to raise the score.

**Key Questions For Authors:**

See weaknesses

**Limitations:**

See weaknesses

**Strengths And Weaknesses:**

### Strength
1. Well-motivated and well-written --> identified an important problem -- random rollout and lack of time sensitive feedback in reward
2. Reward design is interesting, as it take temporal sensitivity into consideration.
3. Extensive experiments across benchmarks and models demonstrates the effectiveness of the proposal TaRO method in VTG tasks.

### Weakness
1. The term "reasoning quality" was mentioned in the paper multiple times, but im not sure if it's being defined properly. It's still a bit vague to me. Do you refer to the length of the reasoning trace? or defined manually by human experts?
2. Line 319, ablation on Time-R1 is good, but the performance improvement might came from the external dense caption generated by Gemini-3-pro. Do you think this is a fair comparison because you are comparing strong caption model with a simple model.
3. Im curious can this method be applied to a broader range of tasks, like general video qa rather than VTG? Intuitively, if a model can do a better VTG, it can perform better on video benchmarks focusing on visual-temporal reasoning.

---

> ### Author Rebuttal · Authors · 2026-03-31
>
> # Q1: The definition of reasoning quality.
>
> Reasoning quality refers to whether the intermediate reasoning process genuinely improves the final grounding performance compared to a direct-prediction baseline. To achieve and optimize this, we designed: (1) **Constructive Reasoning Exploration**, encouraging the model to think with time and selectively use key visual cues;  (2) a **Temporal-Sensitive Reward** to automatically score each path, encouraging the model to think with strict consideration of event temporal sensitivity; and (3) a **Progressive Curriculum** for smooth training transitions, encouraging the model to autonomously generate robust, time-grounded reasoning.
>
> # Q2: Improvement might come from the external dense caption generated by Gemini-3-pro.
>
> (1) As demonstrated in our ablation study (Table 3 of the main text) or the last row of the table below, even without any external captioner, introducing only the Temporal Reward already yields a clear performance boost, improving R1@0.5 from 61.1% to 63.1% on Charades-STA and outperforms the second-best method (Time-R1).
>
> (2) We also replaced Gemini-Pro with the open-source Qwen3.5-9B and Qwen3.5-4B for dense caption generation. Results on Charades-STA are shown below. It yields only marginal performance degradation. This is because our Constructive Reasoning primarily serves to teach the model the pattern of thinking with time and selecting relevant visual cues, rather than requiring flawless, oracle-level descriptions.
>
> | **Method** | **R1@0.3** | **R1@0.5** | **R1@0.7** |
> | --- | --- | --- | --- |
> | Time-R1 | 78.1 | 60.8 | 35.3 |
> | Ours w/ Constructive Reasoning (Gemini-3-Pro) | **79.7** | **64.8** | **38.4** |
> | Ours w/ Constructive Reasoning (Qwen3.5-9B) | 79.5 | 64.2 | 37.9 |
> | Ours w/ Constructive Reasoning (Qwen3.5-4B) | 79.3 | 64.3 | 37.5 |
> | Ours w/o Constructive Reasoning | 78.6 | 63.1 | 36.1 |
>
> (3) We found that Gemini's grounding performance is actually not very good. The table below shows the performance on TVGBench. This confirms that our gains do not come from simply distilling a superior model's grounding ability. Gemini only provides raw, unselected atomic events, while our framework explicitly teaches the model the crucial skill of thinking with time and selecting relevant visual cues.
>
> | **Method** | **R1@0.3** | **R1@0.5** | **R1@0.7** |
> | --- | --- | --- | --- |
> | Gemini | 39.1  | 24.4  | 12.8 |
> | Time-R1 | 41.8  | 29.4  | 16.4 |
> | Ours | 54.6  | 37.8  | 20.0 |
>
> # Q3: Performance on general video QA benchmarks.
>
> To verify this, we evaluated our TaRO model on both general Video-QA benchmarks (TempCompass, MVBench) and a Grounded Video-QA benchmark (NextGQA). The results are presented below. Our method consistently outperforms both the original Qwen2.5-VL-7B base model and the Time-R1 baseline across all evaluated QA metrics. This empirically confirms that by explicitly teaching the model to "think with time" and accurately ground visual cues, our TaRO framework effectively enhances the model's broader video understanding and reasoning capabilities.
>
> **Accuracy on Video-QA benchmarks.**
>
> | **Method** | **TempCompass (Multi-Choice)** | **TempCompass (Caption Matching)** | **MVBench** |
> | --- | --- | --- | --- |
> | Qwen2.5-VL-7B | 69.7 | 81.3 | 64.8 |
> | Time-R1 | 70.9 | 81.3 | 66.0 |
> | Ours | 71.7 | 83.0 | 66.6 |
>
> **Performance on Grounded Video-QA benchmark (NextGQA)**
>
> | **Method** | **Acc@IoP@0.5** | **Acc@GQA** | **mIoP** |
> | --- | --- | --- | --- |
> | Qwen2.5-VL-7B | 72.7 | 42.3 | 54.0 |
> | Time-R1 | 76.9 | 51.1 | 63.6 |
> | Ours | 77.8 | 54.2 | 64.9 |

---

> > ### Author Rebuttal · Reviewer_RHKL · 2026-04-04
> >
> > Thank you for your rebuttal message. I will make my final decision after the discussion period.

---

> > > ### Author Response · Authors · 2026-04-07
> > >
> > > Dear Reviewer RHKL,
> > >
> > > We sincerely thank you for your careful and constructive review, and for acknowledging the motivation, temporal-sensitive reward design, and comprehensive experiments in our work. We are pleased to have addressed your concerns and will include clarifications on the definition of reasoning quality, analysis of different captioners, and discussion of our method’s applicability to general video QA benchmarks in the final version.
> > >
> > > Regards,
> > >
> > > Authors

---

### Official Review · Reviewer_AiHA · 2026-03-13

**Soundness:** 2
**Presentation:** 3
**Significance:** 2
**Originality:** 3
**Overall Recommendation:** 4
**Confidence:** 3

**Summary:**

* The paper looks at an important weakness in RL-based video temporal grounding. In many existing methods, the reasoning text sounds plausible, but it is often quite shallow and not really tied to the actual visual-temporal evidence, so it does not help much for localization.

* To address this, the paper proposes TaRO, with three main parts. First, it replaces random RL rollouts with constructed reasoning traces based on dense video captions, so the model can learn to reason with more concrete timestamped events. Second, it designs a temporal-sensitivity reward: if the model’s confidence drops after shuffling frames near the true event boundary, the reasoning is considered more temporally grounded. Third, the training uses a progressive curriculum, starting from constructed reasoning with behavioral cloning, and then moving to freer RL exploration with the temporal reward.

* Experiments on 4 benchmarks show consistent improvements over previous methods, including Time-R1, especially on smaller models. One result I found quite interesting is that all reasoning traces generated by TaRO contain explicit timestamps, Overall the idea is clear and the empirical results are solid, though some parts of the pipeline feel a bit complex and maybe depend heavily on external caption quality.

**Compliance With Llm Reviewing Policy:**

Affirmed.

**Final Justification:**

As mentioned in the Summary & Strengths And Weaknesses, generally the paper has more good inspiration than some minor points to be revised. I will say it's a weak acc.

**Key Questions For Authors:**

* the reward needs log-probabilities from both the original video and the shuffled one, which basically means two forward passes for each rollout during training. It would be helpful if the authors could report the actual extra training time and memory usage. Without this, it is hard to judge whether the method is practical, espically for larger models.
* the constructive reasoning part depends on Gemini-Pro to produce dense captions, so the method seems to rely on a fairly strong external tool. I wonder how sensitive the final performance is to caption quality. Did the authors try weaker or open-source captioners? If the method drops a lot under those settings, then the general usefulness may be more limited than it first appears.
* all experiments are on relatively short clips. I am not fully convinced the same pipeline would transfer well to long-form videos, where dense captioning is much more expensive and event boundaries can be messier or less clear. Some discussion here would make the paper stronger.
* one concern is whether the model could learn reasoning that is merely highly sensitive to frame shuffling, rather than truly grounded in the correct semantic events. Some extra analysis on this point would be nice, otherwise this reward may be gamed in subtle ways.

**Limitations:**

yes

**Strengths And Weaknesses:**

* The empirical section is pretty solid overall. The ablations are detailed and do a good job separating the effect of each component. I also think the observation that Time-R1’s reasoning brings only very limited gain is a strong and quite honest motivation for this work. The progressive curriculum also makes sense, especially since the constructed rollouts are off-policy in nature, and the use of advantage-weighted behavioral cloning feels reasonable here.

* Presentation: The paper is generally easy to follow, with a clear storyline.

* Focusing on reasoning quality itself, instead of only final grounding accuracy, is a meaningful angle and not much studied yet in this line of work.
* The temporal-sensitivity reward is probly the most original part; using local frame shuffling to test whether reasoning is truly time-grounded is a neat idea. The data efficiency result is also practically interesting.
* The method depends quite a lot on Gemini-Pro dense captions. Also, the extra cost of dual forward passes is not discussed.
* The results are only shown on relatively short-video benchmarks, so scalability to long videos is still unclear.

---

> ### Author Rebuttal · Authors · 2026-03-31
>
> # Q1: Sensitivity to caption quality and open-source captioners.
>
> See **Reviewer qvR8, Q1.**
>
> # Q2: Extra training time and memory for dual log-probs computation.
>
> **Training Time:** Our additional forward pass for shuffled videos increases the per-step training time by only 13.8%. Because GRPO also requires multiple forward passes to generate rollouts and compute $\pi_{\theta old}$, $\pi_{ref}$, $\pi_\theta$ in its optimization objective:
>
> $$J(\theta)=\mathbb{E}[\frac{\pi_\theta}{\pi_{\theta old}}A-\beta D_{KL}(\pi_\theta||\pi_{ref})]$$
>
> As shown in the table below (average time in seconds per step on 8xAscend 910B), our method only adds one extra forward pass for $\pi_{\theta old}$ on shuffled videos. This results in a training time increase of 13.8%, but it also yields substantial gains.
>
> | **Method** | **Rollout Gen.** |  **Forward ( $\pi_{ref}$)** | **Forward ($\pi_{\theta old}$)** | **Foward & Backward ($\pi_\theta$)** | **ALL** | **R1@0.5 on Charades** |
> | --- | --- | --- | --- | --- | --- | --- |
> | Time-R1 | 48.3 | 35.2 | 22.1 | 48.0 | 154.1 | 60.8 |
> | Ours | 49.1 | 35.0 | 40.3 | 50.6 | 175.3 | 64.8 |
>
> **Memory Usage:**  Both our method and the Time-R1 baseline maintain the same peak memory usage of 62.4G during training. This is because the extra forward pass only computes log-probs for the Temporal Reward; gradients are not stored, and intermediates are released immediately.
>
> # Q3: Performance on long-form videos.
>
> We have evaluated four benchmarks in the main text, including QVHighlights, which has the longest average duration at 149s. We further evaluate the zero-shot performance on two long-form benchmarks: **TACoS** (avg. 367s) and **Ego4D NLQ** (avg. 499s).
>
> (1) Using the same Qwen2.5-VL-7B, our method still achieves SOTA and outperforms the second-best baseline, Time-R1.
>
> (2) We additionally compare with the latest Qwen3-VL-8B, which has shown better performance in long-videos than Qwen2.5-VL. Built on top, our method achieves significant improvements (e.g., +13.7% on TACoS and +8.7% on Ego4D NLQ for R1@0.3). This demonstrates that our method is scalable to long-form videos and generalizable to new, stronger baseline architectures.
>
> **TACoS  (avg. 367s)**
>
> | **Method** | **R1@0.3** | **R1@0.5** | **R1@0.7** |
> | --- | --- | --- | --- |
> | VideoChat-R1.5 | 28.6 | 16.2 | 6.17 |
> | Time-R1 | 36.8 | 21.5 | 7.9 |
> | **Ours (Qwen2.5-VL-7B-Instruct )** | **37.9** | **22.5** | **8.6** |
> | Qwen3-VL-8B-Instruct | 38.6 | 30.5 | 20.4 |
> | **Ours (Qwen3-VL-8B-Instruct)** | **52.3** | **36.5** | **22.5** |
>
> **Ego4D NLQ  (avg. 499s)**
>
> | **Method** | **R1@0.3** | **R1@0.5** | **R1@0.7** |
> | --- | --- | --- | --- |
> | VideoChat-R1.5 | 3.84 | 1.78 | 0.86 |
> | Time-R1 | 6.48 | 3.29 | 1.49 |
> | **Ours (Qwen2.5-VL-7B-Instruct )** | **8.19** | **4.55** | **1.93** |
> | Qwen3-VL-8B-Instruct | 5.10 | 2.94 | 1.52 |
> | **Ours (Qwen3-VL-8B-Instruct)** | **13.8** | **7.86** | **4.17** |
>
> # Q4: Whether the model could learn reasoning that is merely sensitive to frame shuffling, rather than truly grounded in the correct semantic events.
>
> (1) Theoretically, shuffled videos are used **only** in a forward pass to compute the reward; parameter updates occur solely on original videos. This prevents the model from learning shortcuts that rely on shuffled-frame positions.
>
> (2) Experimentally, we shuffled frames of non-ground-truth events on Charades-STA. If the model were sensitive only to shuffling, its reasoning would be disrupted. As shown below, performance remains stable, confirming it ignores irrelevant temporal disruptions and grounds on correct events.
>
> (3) We extracted timestamps from the model’s reasoning and computed recall against GT (last row). High recall shows the model relies on key visual cues. Notably, R1@0.3 recall exceeds final prediction accuracy. This may be because intermediate reasoning decomposes composite events into smaller sub-events, yielding higher recall at loose thresholds before synthesizing final predictions.
>
> | **Method** | **R1@0.3** | **R1@0.5** | **R1@0.7** |
> | --- | --- | --- | --- |
> | Time-R1 | 78.2 | 61.1 | 35.2 |
> | Ours | 79.7 | 64.8 | 38.4 |
> | Ours+shuffle | 80.3 | 66.2 | 39.4 |
> | Ours Reasoning Recall | 83.2 | 64.9 | 36.3 |

---

> > ### Author Rebuttal · Reviewer_AiHA · 2026-04-06
> >
> > Thank the authors for the detailed and thoughtful rebuttal, I will maintain my original score.

---

> > > ### Author Response · Authors · 2026-04-07
> > >
> > > Dear Reviewer AiHA,
> > >
> > > We sincerely appreciate your detailed and thoughtful review and the recognition of the clear motivation and solid empirical validation of our approach. We are glad that we have addressed your concerns and will incorporate the impact of different captioners, extra training time, memory usage, and evaluations on longer-form videos in the final version.
> > >
> > > Regards,
> > >
> > > Authors

---

### Decision · Program_Chairs · 2026-04-30

**Decision:**

Accept (regular)

**Comment:**

Recent RL-based methods for video temporal grounding often produce reasoning that appears plausible on the surface but is only weakly grounded in the temporal evidence required for accurate localization. To address this issue, this paper proposes TaRO, a framework that combines constructed reasoning exploration, a temporal-sensitivity reward, and a progressive training curriculum. The reviewers generally agree that the paper is well motivated, clearly presented, and supported by strong empirical results. The main concerns relate to the method’s reliance on external dense captioning, the limited discussion of training overhead, and its unclear scalability to longer videos. Some reviewers also questioned whether the observed gains can be fully attributed to improved reasoning quality itself. Nevertheless, the rebuttal has satisfactorily addressed the major concerns, and the reviewers have given positive overall evaluations. Overall, I recommend Accept.